# ENHANCING THE INDUCTIVE BIASES OF GRAPH NEURAL ODE FOR MODELING PHYSICAL SYSTEMS

**Suresh Bishnoi, Ravinder Bhattoo, Jayadeva, Sayan Ranu, N. M. Anoop Krishnan** [*]
Indian Institute of Technology Delhi, Hauz Khas, New Delhi, India 110016
{srz208500,cez177518,jayadeva,sayanranu,krishnan}@iitd.ac.in

## ABSTRACT

Neural networks with physics-based inductive biases such as Lagrangian neural networks (LNN), and Hamiltonian neural networks (HNN) learn the dynamics of physical systems by encoding strong inductive biases. Alternatively, Neural ODEs with appropriate inductive biases have also been shown to give similar performances. However, these models, when applied to particle-based systems, are transductive in nature and hence, do not generalize to large system sizes. In this paper, we present a graph-based neural ODE, GNODE, to learn the time evolution of dynamical systems. Further, we carefully analyze the role of different inductive biases on the performance of GNODE. We show that similar to LNN and HNN, encoding the constraints explicitly can significantly improve the training efficiency and performance of GNODE significantly. Our experiments also assess the value of additional inductive biases, such as Newton's third law, on the final performance of the model. We demonstrate that inducing these biases can enhance the performance of the model by orders of magnitude in terms of both energy violation and rollout error. Interestingly, we observe that the GNODE trained with the most effective inductive biases, namely MCGNODE, outperforms the graph versions of LNN and HNN, namely, Lagrangian graph networks (LGN) and Hamiltonian graph networks (HGN) in terms of energy violation error by 4 orders of magnitude for a pendulum system, and 2 orders of magnitude for spring systems. These results suggest that NODE-based systems can give competitive performances with energy-conserving neural networks by employing appropriate inductive biases.

## 1 INTRODUCTION AND RELATED WORKS

Learning the dynamics of physical systems is a challenging problem that has relevance in several areas of science and engineering such as astronomy (motion of planetary systems), biology (movement of cells), physics (molecular dynamics), and engineering (mechanics, robotics)(LaValle, 2006; Goldstein, 2011). The dynamics of a system are typically expressed as a differential equation, solutions of which may require the knowledge of abstract quantities such as force, energy, and drag(LaValle, 2006; Zhong et al., 2020; Sanchez-Gonzalez et al., 2020). From an experimental perspective, the real observable for a physical system is its trajectory represented by the position and velocities of the constituent particles. Thus, learning the abstract quantities required to solve the equation, directly from the trajectory, can extremely simplify the problem of learning the dynamics (Finzi et al., 2020).

Infusing physical laws as prior has been shown to improve learning in terms of additional properties such as energy conservation, and symplectic nature (Karniadakis et al., 2021; Lutter et al., 2019; Liu et al., 2021). To this extent, three broad approaches have been proposed, namely, *Lagrangian neural networks* (LNN) (Cranmer et al., 2020a; Finzi et al., 2020; Lutter et al., 2019), *Hamiltonian neural networks* (HNN) (Sanchez-Gonzalez et al., 2019; Greydanus et al., 2019; Zhong et al., 2020; 2021), and *neural ODE* (NODE) (Chen et al., 2018; Gruver et al., 2021). The learning efficiency of LNNs and HNNs is shown to enhance significantly by employing explicit constraints (Finzi et al., 2020) and their inherent structure Zhong et al. (2019). In addition, it has been shown that the superior performance of HNNs and LNNs is mainly due to their second-order bias, and not due to their symplectic or energy conserving bias (Gruver et al., 2021). More specifically, an HNN with separable potential ($V(q)$) and kinetic ($T(q, \dot{q})$) energies is equivalent to a second order NODE of the form $\ddot{q} = \hat{F}(q, \dot{q})$. Thus, a NODE with a second-order bias can give similar performances to

---

[*]SB: School of Interdisciplinary Research, SR: Department of Computer Science, NMAK and RB: Department of Civil Engineering, J: Department of Electrical Engineering, SR and NMAK: Yardi School of Artificial Intelligence (joint appointment).

that of an HNN (Gruver et al., 2021). Recently, a Bayesian-symbolic approach, which assumes the knowledge of Newtonian mechanics along with symbolic features, has been used to learn the dynamics of physical systems Xu et al. (2021).

An important limitation pervading these models is their *transductive* nature; they need to be trained on each system separately before simulating their dynamics. For instance, a NODE trained on 5-pendulum can work only for 5-pendulum. This approach is referred to as transductive. In contrast, an inductive model, such as GNODE, when trained on 5-pendulum "learns" the underlying function governing the dynamics at a node and edge level. Hence, a GNODE, once trained, can perform inference on *any* system size that is significantly smaller or larger than the ground truth as we demonstrate later in § 4. This inductive ability is enabled through Graph neural networks (Scarselli et al., 2008) due to their inherent topology-aware setting, which allows the learned system to naturally generalize. Although the idea of graph-based modeling has been suggested for physical systems (Cranmer et al., 2020b; Greydanus et al., 2019), the inductive biases induced due to different graph structures and their consequences on the dynamics remain poorly explored. In this context, there are a few recent works that aim to develop better representations of physical systems through equivariant graph neural networks Satorras et al. (2021); Han et al. (2022); Huang et al. (2022). However, the present work focuses on the inductive biases for physical systems rather than focusing on better representation learning. Some of the proposed inductive biases may also be a natural consequence of forcing equivariance. For example, enforcing invariance to the Lagrangian w.r.t. space translation and rotation also enforces Newton's third law, which we achieve through an inductive bias (§ 3). In the context of generalizability to unseen systems, there exists work on *few-shot* based on *meta-learning* (Lee et al., 2021). Meta-learning assumes the availability of a limited amount of training data to adapt to an unseen system (task). This is different from our objective where we perform zero-shot generalizability, i.e., inference on an unseen system without any further training. To summarize, our key contributions are as follows:

1. **Topology-aware modeling**, where the physical system is modeled using a graph-based NODE (GNODE), enabling zero-shot generalizability to unseen system sizes.
2. **Decoupling the dynamics and constraints**, where the equation of motion of the GNODE is decoupled into individual terms such as forces, Coriolis-like terms, and explicit constraints.
3. **Decoupling the internal and body forces**, where the internal forces due to the interaction among particles are decoupled from the forces due to external fields.
4. **Newton's third law**, where the forces of interacting particles are enforced to be equal and opposite. We theoretically prove that this inductive bias enforces the conservation of linear momentum, in the absence of an external field, for the predicted trajectory exactly.

We analyze the role of these biases on $n$-pendulum and spring systems. Interestingly, depending on the nature of the system, we show that some of these biases can either significantly improve the performance or have marginal effects. We also show that the final model with the appropriate inductive biases significantly outperforms the simple version of GNODE with no additional inductive biases and even the graph versions of LNN and HNN.

## 2 BACKGROUND ON DYNAMICAL SYSTEMS

A dynamical system can be defined by ordinary differential equations (ODEs) of the form, $\ddot{q} = F(q, \dot{q}, t)$, where $F$ represents the dynamics of the system. Thus, the time evolution of the system can be obtained by integrating this equation of motion. In Lagrangian mechanics, the dynamics of a system are derived based on the Lagrangian of the system, which is defined as (Goldstein, 2011):

$$L(q, \dot{q}, t) = T(q, \dot{q}) - V(q) \tag{1}$$

Once the Lagrangian of the system is computed, the dynamics can be obtained using the Euler-Lagrange (EL) equation as $\frac{d}{dt}(\nabla_{\dot{q}} L) = \nabla_q L$. Note that the EL equation is equivalent to D'Alembert's principle written in terms of the force as $\sum_{i=1}^{n}(N_i - m_i \ddot{q}_i) \cdot \delta q_i = 0$, where $\delta q_i$ represents a virtual displacement consistent with the constraints of the system. Thus, an alternative to the EL equation can be written in terms of the forces and other components as (LaValle, 2006; Murray et al., 2017)

$$M(q)\ddot{q} + C(q, \dot{q})\dot{q} + N(q) + \Upsilon(\dot{q}) + A^T \lambda = \Pi \tag{2}$$

where $M(q)$ represents the mass matrix, $C(q, \dot{q})$ the Coriolis-like forces, $N(q)$ the conservative forces, $\Upsilon(\dot{q})$ represents the non-conservative forces such as drag or friction, and $\Pi$ represents any

external forces. We note that Lagrangian dynamics is traditionally applied in systems with conservative forces, i.e., where a Lagrangian is well-defined. However, Eq. 2 is generic and can be applied to model any dynamical system irrespective of whether the system has a well-defined Lagrangian or not. Examples of systems where a well-defined Lagrangian may not exist include that of a colloidal system undergoing Brownian motion.

The constraints of the system are given by $A(q)\dot{q} = 0$, where $A(q) \in \mathbb{R}^{kD}$ represents $k$ constraints. Thus, $A$ represents the non-normalized basis of constraint forces given by $A = \nabla_q(\phi)$, where $\phi$ is the matrix corresponding to the constraints of the system and $\lambda$ is the Lagrange multiplier corresponding to the relative magnitude of the constraint forces. Note that $M, C$, and $N$ are directly derivable from the Lagrangian of the system $L$ as $M = \nabla_{\dot{q}\dot{q}}L, C = \nabla_{q\dot{q}}L$, and $N = \nabla_q L$. Thus, decoupling the dynamic equation simplifies the representation of the dynamics $F$ with enhanced interpretability. To learn the dynamics, Eq. 2 can be represented as

$$\ddot{q} = M^{-1}\left(\Pi - C\dot{q} - N - \Upsilon - A^T\lambda\right) \tag{3}$$

To solve for $\lambda$, the constraint equation can be differentiated with respect to time to obtain $A\ddot{q} + \dot{A}\dot{q} = 0$. Substituting for $\ddot{q}$ from Eq. 3 and solving for $\lambda$, we get

$$\lambda = (AM^{-1}A^T)^{-1}(AM^{-1}(\Pi - C\dot{q} - N - \Upsilon) + \dot{A}\dot{q}) \tag{4}$$

Accordingly, $\ddot{q}$ can be obtained as

$$\ddot{q} = M^{-1}\left(\Pi - C\dot{q} - N - \Upsilon - A^T(AM^{-1}A^T)^{-1}\left(AM^{-1}(\Pi - C\dot{q} - N - \Upsilon) + \dot{A}\dot{q}\right)\right) \tag{5}$$

## 3  INDUCTIVE BIASES FOR NEURAL ODE

To learn dynamical systems, NODEs parameterize the dynamics $F(q_t, \dot{q}_t)$ using a neural network and learn the approximate function $\hat{F}(q_t, \dot{q}_t)$ by minimizing the loss between the predicted and actual trajectories, that is, $\mathcal{L} = ||q_{t+1} - \hat{q}_{t+1}||$. Thus, a NODE essentially learns a cumulative $\hat{F}$, that consists of the contribution of all the terms contributing to the dynamics. In contrast, in an LNN, the learned $L$ is used in the EL equation to obtain the dynamics. Similarly, in LNN with explicit constraints, the learned Lagrangian is used to obtain the individual terms in the Eq. 5 by differentiating the LNN, which is then substituted to obtain the acceleration. However, this approach is transductive in nature. To overcome this challenge, we present a graph-based version of the NODE namely, GNODE, which models the physical system as a graph. We show, that GNODE can generalize to arbitrary system size after learning the dynamics $\hat{F}$.

### 3.1  GRAPH NEURAL ODE (GNODE) FOR PHYSICAL SYSTEMS

In this section, we describe the architecture of GNODE. The graph topology is used to learn the approximate dynamics $\hat{F}$ by minimizing the loss on the predicted positions.

**Graph structure.** An $n$-particle system is represented as an undirected graph $\mathcal{G} = \{\mathcal{V}, \mathcal{E}\}$, where nodes represent particles and edges represent the connections or interactions between them. For instance, in a pendulum or springs system, the nodes correspond to bobs or balls, respectively, and the edges correspond to the bars or springs, respectively.

**Input features.** Each node is characterized by the features of the particles themselves, namely, the particle *type* ($t$), *position* ($q_i$), and *velocity* ($\dot{q}_i$). The *type* distinguishes particles of differing characteristics, for instance, balls or bobs with different masses. Further, each edge is represented by the edge features $w_{ij} = (q_i - q_j)$, which represents the relative displacement of the nodes connected by the given edge.

**Neural architecture:** Here, we provide a summary of the various neural layers within GNODE (see a detailed description in App. A and ). Fig. 8 in the appendix provides a pictorial description. In the *pre-processing* layer, we construct a dense vector representation for each node $v_i$ and edge $e_{ij}$ using MLPs. In many cases, internal forces in a system that govern the dynamics are closely dependent on the topology of the structure. We model this through a *deep, message-passing* GNN. Let $\mathbf{z}_i = \mathbf{h}_i^L$ denote the final representation constructed by the GNN over each node. These representations are passed through another MLP to predict the acceleration of each node.

**Trajectory prediction and training.** Based on the predicted $\ddot{q}$, the positions and velocities are predicted using the *velocity Verlet* integration. The loss function of GNODE is computed by using

the predicted and actual accelerations at timesteps $2, 3, \ldots, \mathcal{T}$ in a trajectory $\mathbb{T}$, which is then back-propagated to train the MLPs. Specifically, the loss function is as follows.

$$\mathcal{L} = \frac{1}{n} \left( \sum_{i=1}^{n} \sum_{t=2}^{\mathcal{T}} \left( \ddot{q}_i^{\mathbb{T},t} - \hat{\ddot{q}}_i^{\mathbb{T},t} \right)^2 \right) \tag{6}$$

Here, $(\hat{\ddot{q}}_i^{\mathbb{T},t})$ is the predicted acceleration for the $i^{th}$ particle in trajectory $\mathbb{T}$ at time $t$ and $\ddot{q}_i^{\mathbb{T},t}$ is the true acceleration. $\mathbb{T}$ denotes a trajectory from $\mathfrak{T}$, the set of training trajectories. Note that the accelerations are computed directly from the ground truth trajectory using the Verlet algorithm:

$$\ddot{q}(t) = \frac{1}{(\Delta t)^2} [q(t + \Delta t) + q(t - \Delta t) - 2q(t)] \tag{7}$$

Since the integration of the equations of motion for the predicted trajectory is also performed using the same algorithm as $q(t + \Delta t) = 2q(t) - q(t - \Delta t) + \ddot{q}(\Delta t)^2$, equivalent to training from trajectory/positions. It should be noted that the training approach presented here may lead to learning the dynamics as dictated by the Verlet integrator. Thus, the learned dynamics may not represent the "true" dynamics of the system, but one that is optimal for the Verlet integrator[1].

GNODE learns Newtonian dynamics directly without decoupling the constraints and other components governing the dynamics from the equation $\ddot{q} = \hat{F}(q, \dot{q}, t)$. Next, we analyze the effect of providing the constraints explicitly in the form of inductive biases and enhance the performance of GNODE. In the subsequent sections, we iteratively stack inductive biases on top of GNODE. Thus, the final proposed architecture, MCGNODE, is the eventual model that considers all biases. The detailed differences between all the graph architectures proposed in the work are included in App. K and Fig. 23.

### 3.2 DECOUPLING THE DYNAMICS AND CONSTRAINTS (CGNODE).

To this extent, while keeping the graph architecture exactly the same, we use Eq. 5 to compute the acceleration of the system. In this case, the output of GNODE is $N_i = \texttt{squareplus}(\texttt{MLP}_\mathcal{V}(\mathbf{z}_i))$ instead of $\ddot{q}_i$, where $N_i$ is the conservative force on particle $i$ (refer Eq. 2). Then, the acceleration is predicted using Eq. 5 with explicit constraints. This approach, termed as *constrained* GNODE (CGNODE), enables us to isolate the effect of imposing constraints explicitly on the learning, and the performance of GNODE. It is worth noting that for particle systems in Cartesian coordinates, the mass matrix is a diagonal one, with the diagonal entries, $m_{ii}$s, representing the mass of the $i^{th}$ particle. Thus, the $m_{ii}$s are learned as trainable parameters. As a consequence of this design, the graph directly predicts the $N_i$ of each particle. Note that in the case of systems with drag, the graph would contain a cumulative of $N_i$ and $\Upsilon_i$. Examples of such systems are demonstrated later (see App. E).

### 3.3 DECOUPLING THE INTERNAL AND BODY FORCES (CDGNODE)

In a physical system, internal forces arise due to interactions between constituent particles. On the contrary, the body forces are due to the interaction of each particle with an external field such as gravity or electromagnetic field. While internal forces are closely related to the system's topology, the effect of external fields on each particle is independent of the topology of the structure. In addition, while the internal forces are a function of the relative displacement between the particles, the external forces due to fields depend on the actual position of the particle.

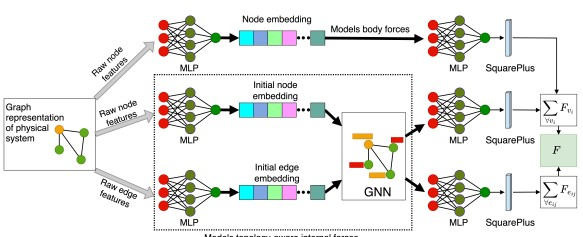

Figure 1: Architecture of CDGNODE (see App. K).

To decouple the internal and body forces, we modify the graph architecture of the GNODE. Specifically, the node inputs are divided into local and global features. Only the local features are involved in the message passing. The global features are passed through an MLP to obtain an embedding, which is concatenated with the node embedding obtained after message passing. This updated embedding is passed through an MLP to obtain the particle level force, $N_i$. Here, the local node

---

[1]See App. A for details on why we use Verlet integrator

features are the particle types and the global features are position and velocity. Note that in the case of systems with drag, the $\Upsilon_i$ can be learned using a separate node MLP using particle type and velocity as global input features. This architecture is termed as *constrained and decoupled* GNODE (CDGNODE). The architecture of CDGNODE is provided in Fig. 1.

### 3.4 NEWTON'S THIRD LAW (MCGNODE)

According to Newton's third law, *"every action has an equal and opposite reaction"*. In particular, the internal forces on each particle exerted by the neighboring particles should be equal and opposite so that the total system is in equilibrium. That is, $f_{ij} = -f_{ji}$, where $f_{ij}$ represents the force on particle $i$ due to its neighbor $j$. Note that this is a stronger requirement than the EL equation itself, i.e., $\frac{d}{dt}\left(\nabla_{\dot{q}_i}L\right) = \nabla_{q_i}L$. EL equation enforces that the rate of change of momentum of each particle is equal to the total force on the particle, $N_i = \sum_{j=1}^{n_i} f_{ij}$, where $n_i$ is the number of neighbors of $i$ — it does not enforce any restrictions on the individual component forces, $f_{ij}$ from the neighbors that add to this force. Thus, although the total force on each particle may be learned by the GNODE, the individual components may not be learned correctly. We show that GNODE with an additional inductive bias enforcing Newton's third law, namely *momentum conserving* GNODE (MCGNODE), conserves the linear momentum of the learned dynamics exactly.

**Theorem 1** *In the absence of an external field,* MCGNODE *exactly conserves the momentum.*

Proof of the theorem is provided in Appendix B. To enforce this bias, we modify the graph architecture as follows. In particular, the output of the graph after the message passing is taken from the edge embedding instead of taking from the node embedding. The final edge embedding is obtained after the message passing is passed through an MLP to obtain the force $f_{ij}$. Then the nodal force is computed as the sum of all the forces from the edges incident on the node. Note that while doing the sum, the symmetry condition $f_{ij} = -f_{ji}$ is enforced. Further, the global forces coming from any external field are added separately to obtain the total nodal forces. Finally, the parameterized masses are learned as the masses in a particle system in Cartesian coordinates are diagonal in nature.

## 4 EMPIRICAL EVALUATION

In this section, we benchmark GNODE and the impact of the inductive biases. Our implementation is available at https://anonymous.4open.science/r/graph_neural_ODE-8B3D. Details of the simulation environment are provided in App. I.

### 4.1 EXPERIMENTAL SETUP

• **Baselines:** To compare the performance of GNODE and its associated versions MCGNODE, CDGNODE, and CGNODE, we compare with graph-based physics-informed neural networks, namely, (1) Lagrangian graph network (LGN) (Cranmer et al., 2020a), and (2) Hamiltonian graph network (Sanchez-Gonzalez et al., 2019). Since the exact architecture and details of the graph architecture are not provided in the original works, a full graph architecture (Cranmer et al., 2020b; Sanchez-Gonzalez et al., 2020) is used for modeling LGN and HGN[2]. Since Cartesian coordinates are used, the parameterized masses are learned as a diagonal matrix as performed in (Finzi et al., 2020). We note that while in GNODE, the predicted acceleration is the direct output of the graph neural network, in LGN (or HGN) the output is the Lagrangian (or Hamiltonian) and further differentiation of the Lagrangian (or Hamiltonian) provides acceleration. These additional computational steps make LGN or HGN computationally expensive. We empirically establish this in App. F. This result is consistent with earlier results that note neural ODEs (NODE) as being more efficient and generalizable than Hamiltonian neural networks if appropriate inductive biases are injected Gruver et al. (2021). Thus, we also compare with NODE (Gruver et al., 2021). Finally, we also compare with SymODE-Net (Zhong et al., 2019) (App. H) and graph network simulator (GNS) (App. L).

• **Datasets and systems:** To compare the performance GNODEs with different inductive biases and with the baseline models, we selected systems on which NODEs are shown to give good performance (Zhong et al., 2021; Finzi et al., 2020), *viz*, $n$-pendulums and springs, where $n = (3, 4, 5)$. Further, to evaluate the zero-shot generalizability of GNODE to large-scale unseen systems, we simulate 5, 20, and 50-link spring systems, and 5-, 10-, and 20-link pendulum systems. The details of the experimental systems are given in Appendix. The training data is generated for systems without drag for each particle. The timestep used for the forward simulation of the pendulum system is $10^{-5}s$ with the data collected every 1000 timesteps and for the spring system is $10^{-3}s$ with the data collected every 100 timesteps. The detailed data-generation procedure is given in App. J.

---

[2]We also evaluated the performance of LGN and HGN on the GNN of GNODE and observe that the performance is inferior. Details are provided in App. C

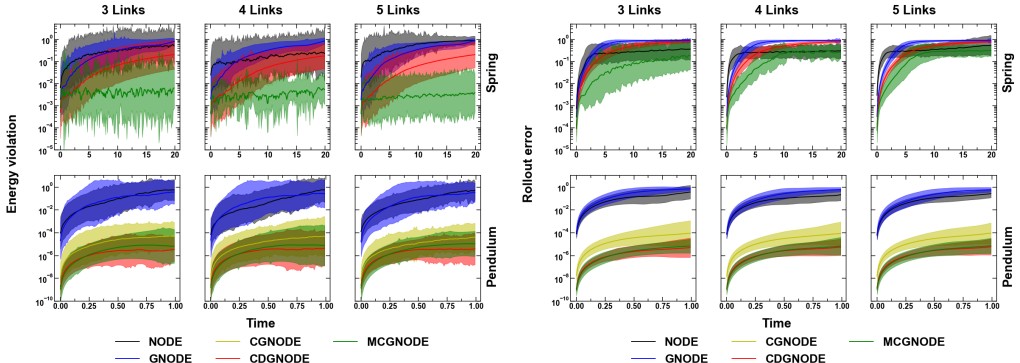

Figure 2: Comparison of the energy violation and rollout error between NODE, GNODE, CGNODE, CDGNODE, MCGNODE for $n$-pendulum and spring systems with $n = 3, 4, 5$ links. For the pendulum system, the model is trained on the 5-pendulum system and for the spring system, the model is trained on the 5-spring system and evaluated on all other systems. The shaded region shows the 95% confidence interval from 100 initial conditions.

• **Evaluation Metric:** Following the work of (Finzi et al., 2020), we evaluate performance by computing the following three error metrics, namely, (1) the relative error in the trajectory, defined as the *rollout error, (RE(t))*, (2) *Energy violation error (EE(t))*, (3) the relative error in *momentum conservation, (ME(t))*, given by: $RE(t) = \frac{||\hat{q}(t) - q(t)||_2}{||\hat{q}(t)||_2 + ||q(t)||_2}$; $EE(t) = \frac{||\hat{\mathcal{H}} - \mathcal{H}||_2}{(||\hat{\mathcal{H}}||_2 + ||\mathcal{H}||_2)}$; $ME(t) = \frac{||\hat{\mathcal{M}} - \mathcal{M}||_2}{||\hat{\mathcal{M}}||_2 + ||\mathcal{M}||_2}$ Note that all the variables with a hat, for example, $\hat{x}$, represent the predicted values based on the trained model, and the variables without the hat, that is $x$, represent the ground truth.

• **Model architecture and training setup:** For all versions of GNODE, LGN, and HGN, the MLPs are two-layers deep. We use 10000 data points generated from 100 trajectories to train all the models. This dataset is divided randomly in a 75:25 ratio as a training and validation set. Detailed training procedures and hyper-parameters are provided in App. J. All models were trained till the decrease in loss saturates to less than 0.001 over 100 epochs. The model performance is evaluated on a forward trajectory, a task it was not explicitly trained for, of $10s$ in the case of the pendulum and $20s$ in the case of spring. Note that this trajectory is 2-3 orders of magnitude larger than the training trajectories from which the training data has been sampled. The dynamics of the $n$-body system are known to be chaotic for $n \geq 2$. Hence, all the results are averaged over trajectories generated from 100 different initial conditions.

## 4.2 EVALUATION OF INDUCTIVE BIASES IN GNODE

• **Performance of (GNODE).** Fig. 2 shows the performance of GNODE on the pendulum and spring systems of sizes 3 to 5. The GNODE for the pendulum is trained on a 5-pendulum system and that of the spring is trained on a 5-spring system. Hence, the 3- and 4-pendulum systems and 3- and 4-spring systems are *unseen*, and consequently, allow us to evaluate zero-shot generalizability. We observe that energy violation and rollout errors of GNODE for the forward simulations are comparable, both in terms of the value and trend with respect to time, across all of the systems. Furthermore, when we compare with NODE, which has been trained on each system separately, while the errors are comparable in 5-pendulum, GNODE is better in 5-spring. It is worth highlighting that NODE has an unfair advantage on 5-pendulum since it is trained and tested on the same system, while 5-pendulum is unseen for GNODE (trained on 5-pendulum). This confirms the generalizability of GNODE to simulate the dynamics of unseen systems of arbitrary sizes. More experiments comparing NODE with GNODE to further highlight the benefit of topology-aware modeling are provided in App.D.

• **GNODE with explicit constraints (CGNODE).** Next, we analyze the effect of incorporating the constraints explicitly in GNODE. To this extent, we train the 5-pendulum system using CGNODE model with acceleration computed using Equation 5. Further, we evaluate the performance of the model on 4- and 5-pendulum systems. Note that the spring system is an unconstrained system and hence CGNODE is equivalent to GNODE for this system. We observe that (see Fig. 2) CGNODE exhibits ∼4 orders of magnitude superior performance both in terms of energy violation and rollout error. This confirms that providing explicit constraints significantly improves both the learning efficiency and the predictive dynamics of GNODE models where explicit constraints are present, for example, pendulum systems, chains, or rigid bodies.

• **Decoupling internal and body forces (CDGNODE).** Now, we analyze the effect of decoupling local and global features in the GNODE by employing the CDGNODE architecture. It is worth

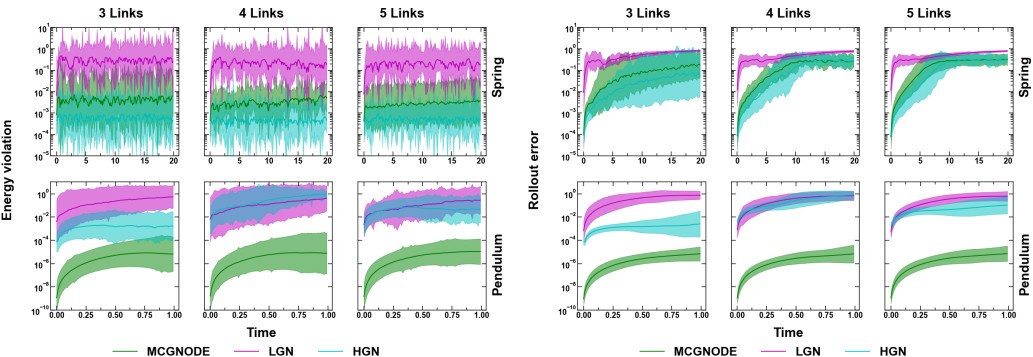

Figure 3: Comparison of the energy violation and rollout error between GNODE, LGN, and HGN for $n$-pendulum and spring systems with $n = 3, 4, 5$ links. For both the pendulum and spring systems, the model is trained on a 5-link structure and evaluated on all other systems. The shaded region shows the 95% confidence interval from 100 initial conditions.

noting that the decoupling of the local and global features makes the GNODE model parsimonious with only the relative displacement and node type given as input features for message passing to the graph architecture. In other words, the number of features given to the graph architecture is lower in CDGNODE in GNODE. Interestingly, we observe that the CDGNODE performs better than CGNODE and GNODE by ∼1 order of magnitude for pendulum and spring systems, respectively.

• **Newton's third law (MCGNODE).** Finally, we empirically evaluate the effect of enforcing Newton's third law, that is, $f_{ij} = -f_{ji}$ on the graph using the graph architecture namely, MCGNODE. Figure 2 shows the performance of MCGNODE for both pendulum and spring systems. Note that it was theoretically demonstrated that this inductive bias enforces the exact conservation of momentum, in the absence of an external field. Figure 4 shows the $ME$ of MCGNODE in comparison to NODE, GNODE, CGNODE, CDGNODE, and MCGNODE for pendulum and spring systems. We observe that in the spring system, the $ME$ of MCGNODE is zero. In the case of a pendulum, although lower than NODE, GNODE, and CGNODE, the $ME$ of MCGNODE is comparable to that of CDGNODE. Correspondingly, MCGNODE exhibits improved performance for the spring system in comparison to other models. However, in the pendulum system, we observe that the performance of MCGNODE is similar to that of CDGNODE. This is because, in spring systems, the internal forces between the balls govern the dynamics of the system, whereas in pendulum the dynamics are governed mainly by the force exerted on the bobs by the external gravitational field. Interestingly, we observe that the energy violation error of MCGNODE remains fairly constant, while this error diverges for other models. This confirms that strong enforcement of momentum conservation in turn ensures that the energy violation error stays constant over time, ensuring stable long-term equilibrium dynamics of the system. In addition, MCGNODE exhibits significantly improved training efficiency than NODE (see App.F). Finally, we also evaluate the ability of MCGNODE to learn the dynamics of systems with drag and external forces (see App. E). We observe that MCGNODE clearly outperforms NODE in this case as well. MCGNODE will, thus, be particularly useful for the statistical analysis of chaotic systems where multiple divergent trajectories correspond to the same energy state.

### 4.3 COMPARISON WITH BASELINES

Now, we compare the GNODE that exhibited the best performance, namely, MCGNODE with the baselines namely, LGN and HGN. Note that both LGN and HGN are energy-conserving graph neural networks, which are expected to give superior performance in predicting the trajectories of dynamical systems (Sanchez-Gonzalez et al., 2019). Especially, due to their energy-conserving nature, these models are expected to have lower energy violation error than other models, where energy conservation bias is not explicitly enforced. Figure 3 shows the energy violation and rollout error of GNODE, LGN, and HGN trained on a 5-link system and tested on 3, 4, 5-link pendulum and spring systems. Interestingly, we observe that MCGNODE outperforms both LGN and HGN in terms of rollout error and energy error for both spring and pendulum systems. Specifically, in the case of pendulum systems, we observe that the energy violation and rollout errors are ∼ 5 orders of magnitude lower in the case of MCGNODE in comparison to LGN and HGN. Further, we compare the $ME$ of MCGNODE with LGN and HGN (see Fig. 4 right palette). We observe in both pendulum and spring systems that the $ME$ of MCGNODE is significantly lower than that of LGN and HGN. Thus, in the case of spring systems, although the performance of HGN and MCGNODE are comparable for both $RE$ and $EE$, we observe that MCGNODE exhibits zero $ME$, while in HGN and LGN $ME$

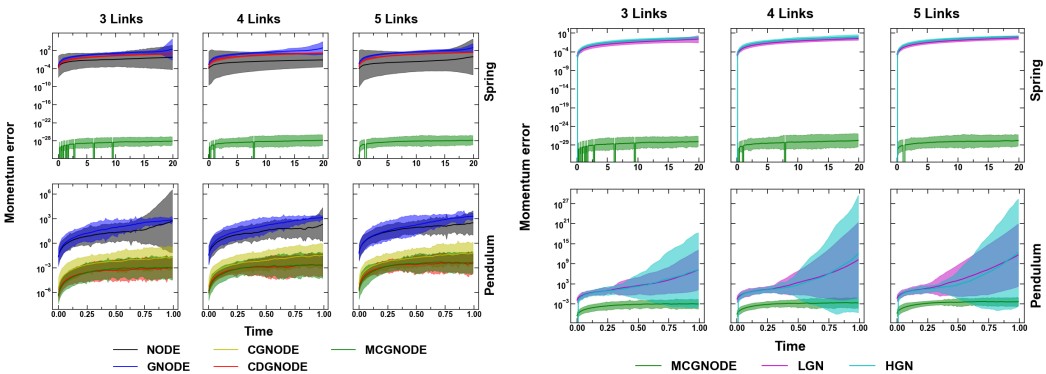

Figure 4: Momentum error ($ME$) of NODE, GNODE, CGNODE, CDGNODE, and MCGNODE (top palette), and LGN, HGN, MCGNODE (bottom palette) for spring (top row of both the palettes) and pendulum (bottom row of both the palettes) systems with 3, 4, and 5 links. The shaded region represents the 95% confidence interval based on 100 forward simulations.

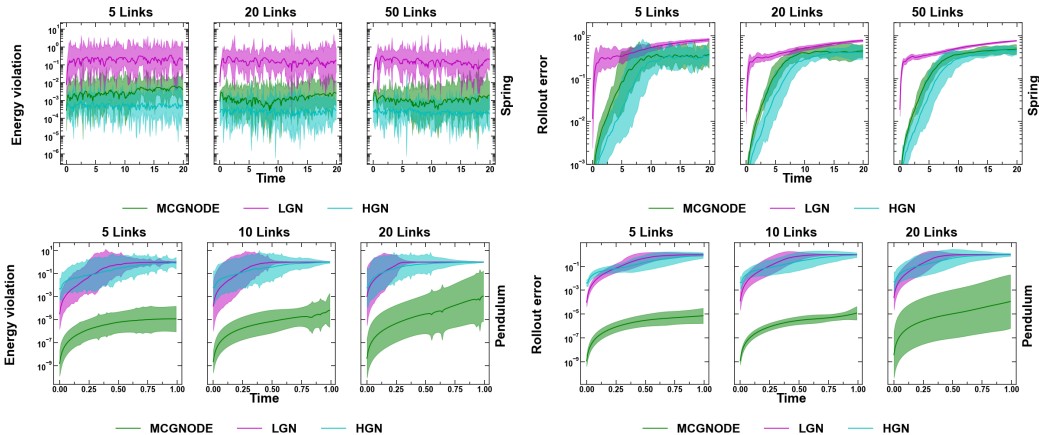

Figure 5: Energy violation and rollout error between GNODE, LGN, and HGN for spring systems with $5, 20, 50$ links and pendulum system with $5, 10, 20$ links. Note that for all the systems, GNODE, LGN, and HGN are trained on 5-pendulum and 5-spring systems and tested on others. The shaded region represents the $95\%$ confidence interval based on 100 forward simulations.

diverges. The superior performance of MCGNODE could be attributed to the carefully constructed inductive biases, namely, (i) explicit constraints—improves data efficiency and reduces the error in pendulum systems, (ii) decoupling internal and body forces—leads to superior performance in pendulum systems which act under gravity, (iii) momentum conservation—leads to superior performance in spring systems by enforcing an additional conservation law strongly. We note that these inductive biases may also be adapted suitably for LGN and HGN and thereby improve their performance. Thus, the contribution of our work is not the neural architecture in isolation, but the core design principles that are generic enough to be abstracted beyond neural ODEs.

## 4.4 GENERALIZABILITY

Now, we focus on the zero-shot generalizability of GNODE, LGN, and HGN to large systems that are unseen by the model. To this extent, we model 5-, 10-, and 20-link pendulum systems and 5-, 20-, and 50-link spring systems. Note that all the models are trained on the 5-link systems. In congruence with the comparison earlier, we observe that MCGNODE performs better than both LGN and HGN when tested on systems with a large number of links. However, we observe that the variation in the error represented by the 95% confidence interval increases with an increase in the number of particles. This suggests that the increase in the number of particles essentially results in increased noise in the prediction of the forces. However, the error in the MCGNODE is significantly lower than the error in LGN and HGN for both pendulum and spring systems, as confirmed by the 95% confidence interval, which could again be attributed to the three inductive biases infused in MCGNODE, that are absent in LGN and HGN. This suggests that MCGNODE can generalize to larger system sizes with increased accuracy.

## 4.5 LEARNING THE DYNAMICS OF 3D SOLID

To demonstrate the applicability of MCGNODE to learn the dynamics of complex systems, we consider a challenging problem, that is, inferring the dynamics of an elastically deformable body. Specifically, a $5 \times 5 \times 5$ solid cube, discretized into 125 particles, is used for simulating the ground truth. A 3D solid system is simulated using the peridynamics framework. The system is

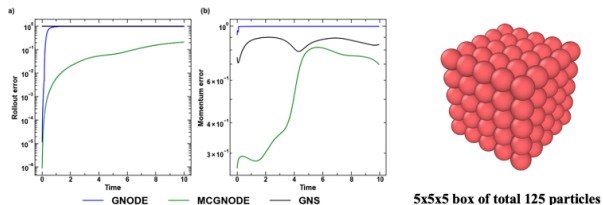

Figure 6: $(RE)$ and $(ME)$ of GNODE, MCGNODE and GNS for particle-based system. Simulation is performed using peridynamics on a 3D box of 125 particles ($5 \times 5 \times 5$).

compressed isotropically and then released to simulate the dynamics of the solid, that is, the contraction and expansion in 3D. We observe that MCGNODE performs inference much superior to GNODE and GNS (see Fig. 6and App. L). It should also be noted that although the momentum error in MCGNODE seems high, this is due to the fact that the actual momentum in these systems is significantly low. This is demonstrated by plotting the absolute momentum error in these systems which in MCGNODE is close to zero, while in GNODE it explodes (see Fig. 21). These results suggest that the MCGNODE could be applied to much more complex systems such as solids or other particle-based systems where momentum conservation is equally important as energy conservation.

## 4.6 ROBUSTNESS TO NOISE

Finally, we show the robustness to noise of the MCGNODE in comparison to GNODE, and additional baselines, namely, HGN and LGN. Fig. 7 shows the performance of these models on 5-spring and 5-pendulum systems for varying noise values. Note that the noise is given in terms of % standard deviation of the distribution of the original output values. We observe that MCGNODE is robust against noise in comparison to other models.

## 5 CONCLUSIONS

In this work, we carefully analyze various inductive biases that can affect the performance of a GNODE. Specifically, we analyze the effect of a graph architecture, explicit constraints, decoupling global and local features that makes the model parsimonious, and Newton's third law that enforces a stronger constraint on the conservation of momentum. We observe that inducing appropriate inductive biases results in an enhancement in the performance of GNODE by ∼2-3 orders magnitude for spring systems and ∼5-6 orders magnitude for pendulum systems in terms of energy violation error. Further, MCGNODE outperforms the graph versions of both Lagrangian and Hamiltonian neural networks, namely, LGN and HGN. Altogether, the present work suggests that the performance of GNODE-based models can be enhanced and even made superior to energy-conserving neural networks by inducing the appropriate inductive biases.

**Limitations and Future Works:** The present work focuses on particle-based systems such as pendulum and spring systems. While these simple systems can be used to evaluate model performances, real life is more complex in nature with rigid body motions, contacts, and collisions. Extending GNODE to address these problems can enable the modeling of realistic systems. A more complex set of systems involve plastically deformable systems such as steel rod, and viscoelastic materials where the deformation depends on the strain rate of materials. Further, instead of explicitly providing the constraints, learning the constraints directly from the trajectory can enhance the application of the models to unseen systems. Note that the present work is limited to physical systems. A similar approach could be employed in other dynamical systems as well. From the neural modeling perspective, the GNODE is essentially a graph convolutional network. Using *attention* over each message passing head may allow us to better differentiate neighbors based on the strength of their interactions. We hope to pursue these questions in the future.

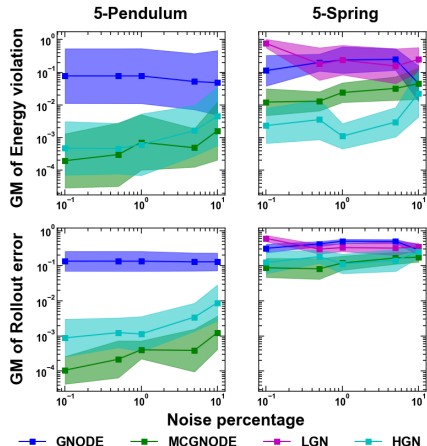

Figure 7: Impact of noise on performance of GNODE, MCGNODE, LGN, and HGN. Note that the noise is given in terms of % standard deviation of the distribution of the original output values.

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

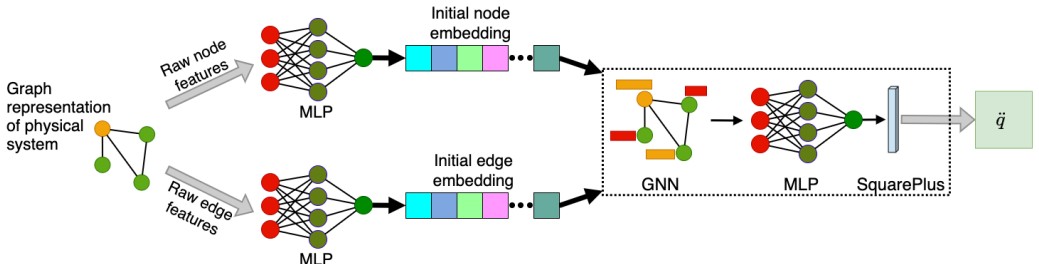

Figure 8: Architecture of GNODE.

## A   DETAILS OF GNODE

The architecture of GNODE is depicted in Fig. 8. GNODE constitutes of the following neural layers.
**Pre-Processing.** In the pre-processing layer, we construct a dense vector representation for each node $v_i$ and edge $e_{ij}$ using $\texttt{MLP}_{em}$ as:

$$\mathbf{h}_i^0 = \texttt{squareplus}(\texttt{MLP}_{em}(\texttt{one-hot}(t_i), q_i, \dot{q}_i)) \tag{8}$$

$$\mathbf{h}_{ij}^0 = \texttt{squareplus}(\texttt{MLP}_{em}(w_{ij})) \tag{9}$$

$\texttt{squareplus}$ is an activation function. Note that the $\texttt{MLP}_{em}$ corresponding to the node and edge embedding functions are parameterized with different weights. Here, for the sake of brevity, we simply mention them as $\texttt{MLP}_{em}$.
**Acceleration prediction.** In many cases, internal forces in a system that govern the dynamics are closely dependent on the topology of the structure. To capture this information, we employ multiple layers of *message-passing* between the nodes and edges. In the $l^{th}$ layer of message passing, the node embedding is updated as:

$$\mathbf{h}_i^{l+1} = \texttt{squareplus}\left(\mathbf{h}_i^l + \sum_{j \in \mathcal{N}_i} \mathbf{W}_{\mathcal{V}}^l \cdot \left(\mathbf{h}_j^l || \mathbf{h}_{ij}^l\right)\right) \tag{10}$$

where, $\mathcal{N}_i = \{v_j \in \mathcal{V} \mid e_{ij} \in \mathcal{E}\}$ are the neighbors of $v_i$. $\mathbf{W}_{\mathcal{V}}^l$ is a layer-specific learnable weight matrix. $\mathbf{h}_{ij}^l$ represents the embedding of incoming edge $e_{ij}$ on $v_i$ in the $l^{th}$ layer, which is computed as follows.

$$\mathbf{h}_{ij}^{l+1} = \texttt{squareplus}\left(\mathbf{h}_{ij}^l + \mathbf{W}_{\mathcal{E}}^l \cdot \left(\mathbf{h}_i^l || \mathbf{h}_j^l\right)\right) \tag{11}$$

Similar to $\mathbf{W}_{\mathcal{V}}^l$, $\mathbf{W}_{\mathcal{E}}^l$ is a layer-specific learnable weight matrix specific to the edge set. The message passing is performed over $L$ layers, where $L$ is a hyper-parameter. The final node and edge representations in the $L^{th}$ layer are denoted as $\mathbf{z}_i = \mathbf{h}_i^L$ and $\mathbf{z}_{ij} = \mathbf{h}_{ij}^L$ respectively. Finally, the acceleration of the particle $\ddot{q}_i$ is predicted as:

$$\ddot{q}_i = \texttt{squareplus}(\texttt{MLP}_{\mathcal{V}}(\mathbf{z}_i)) \tag{12}$$

**Why Verlet integrator:** The canonical coordinates employed in Newtonian mechanics are the position $(q)$ and velocity $(\dot{q})$ (or momentum) as these quantities are the direct observables. Verlet algorithm is obtained by substracting the Taylor series expansion of $r(t + \Delta t)$ with $r(t - \Delta t)$. The expression thus obtained is time-reversible and has error to the order of $(\mathcal{O}(\Delta t^4))$. Due to the time-reversible nature, Verlet algorithm is energy conserving (symplectic), and hence performs better than other traditional integrators such as RK4 in predicting the long term trajectory of dynamical systems without drift.

## B   PROOF OF THEOREM 1

PROOF. Consider a system without any external field where the dynamics is governed by internal forces, for instance, a $n$-spring system. The conservation of linear momentum for this system implies

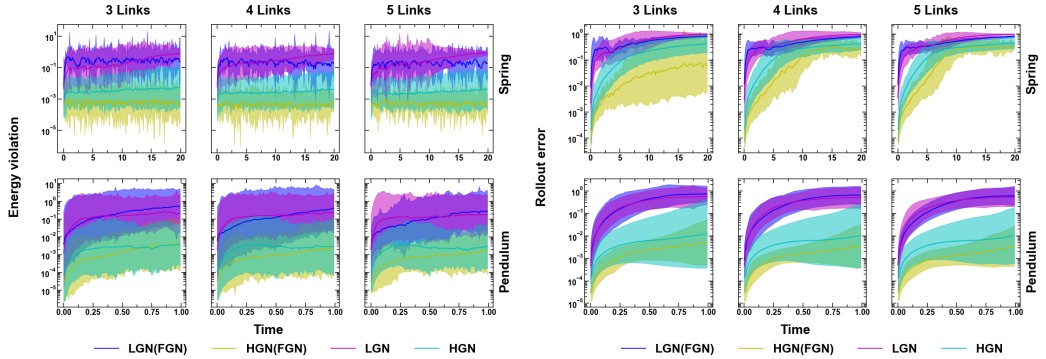

Figure 9: comparison of LGN and HGN with the full graph architecture.

that $\sum_{i=1}^{n} \frac{d}{dt} \left( \hat{m}_{ii} \hat{\dot{q}}_i \right) = 0$, where $\hat{\dot{q}}_i$ represents the predicted velocity, $\hat{m}_{ii}$ represents the learned mass of the particle $i$, and $\dot{q}_i$ represents the ground truth velocity. Assume the error in the rate of change of momentum of the predicted trajectory to be $\hat{\varepsilon}_t$. To prove that MCGNODE conserves the momentum exactly, it is sufficient to prove that $\hat{\varepsilon}_t = 0$. Consider,

$$\hat{\varepsilon}_t = \sum_{i=1}^{n} \hat{\varepsilon}_{i,t} = \sum_{i=1}^{n} \frac{d}{dt} \left( \hat{m}_{ii} \hat{\dot{q}}_i - m_{ii} \dot{q}_i \right) = \sum_{i=1}^{n} (\hat{N}_i - N_i) = \sum_{i=1}^{n} \hat{N}_i - \sum_{i=1}^{n} N_i \tag{13}$$

Note that in equation 3, all terms except $N_i$ are either known or are zero. Thus, without loss of generality, $\sum_{i=1}^{n} N_i = 0$, in the absence of an external field. Hence,

$$\hat{\varepsilon}_t = \sum_{i=1}^{n} \hat{\varepsilon}_{i,t} = \sum_{i=1}^{n} \hat{N}_i = \sum_{i=1}^{n} \sum_{j=1}^{n_i} f_{ij} = \sum_{i=1}^{n} \sum_{j=1}^{n_i} \frac{(f_{ij} + f_{ji})}{2} \tag{14}$$

The inductive bias $f_{ij} = -f_{ji}$ implies that $\frac{(f_{ij} + f_{ji})}{2} = 0$. Thus,

$$\sum_{i=1}^{n} \frac{d}{dt} \left( m_{ii} \hat{\dot{q}}_i \right) = \hat{\varepsilon}_t = 0 \qquad \square$$

## C  PERFORMANCE OF LGN AND HGN WHEN IMPLEMENT ON THE GNN OF GNODE.

In Fig. 9, we compare the performance of LGN and HGN implemented on the full graph network (Cranmer et al., 2020b; Sanchez-Gonzalez et al., 2020) against the versions implemented on the GNN used in GNODE. As visible, the full graph network version provides superior performance.

## D  ZERO-SHOT GENERALIZABILITY: NODE VS GNODE VS MCGNODE

To showcase zero-shot generalizability, we compare the performance of NODE, GNODE, MCGNODE in Figure 10. In the case of NODE, the system is trained and evaluated on each of the pendulum and spring systems separately. In the case of GNODE and MCGNODE, the training is performed on 5-spring and 5-pendulum system and evaluated on all other systems. First, we observe that the performance of NODE and GNODE are comparable for both rollout and energy error. This suggests that the performance of GNODE, with no additional inductive bias, is comparable to that of NODE with the additional advantage that GNODE can generalize to any system size. Now, we compare the performance of NODE with MCGNODE, which has all the additional inductive biases discussed earlier. We observe that MCGNODE performs significantly better than NODE, as expected. It is worth noting that while the inductive bias of explicit constraints can be infused in NODE in a similar fashion as in GNODE, other inductive biases such as decoupling the global and local features and Newton's third law are unique to the graph structure. These inductive biases cannot be infused in a NODE structure. This, in turn, demonstrates the superiority of GNODE which allows additional inductive biases due to the explicit representation of the topology through the graph structure.

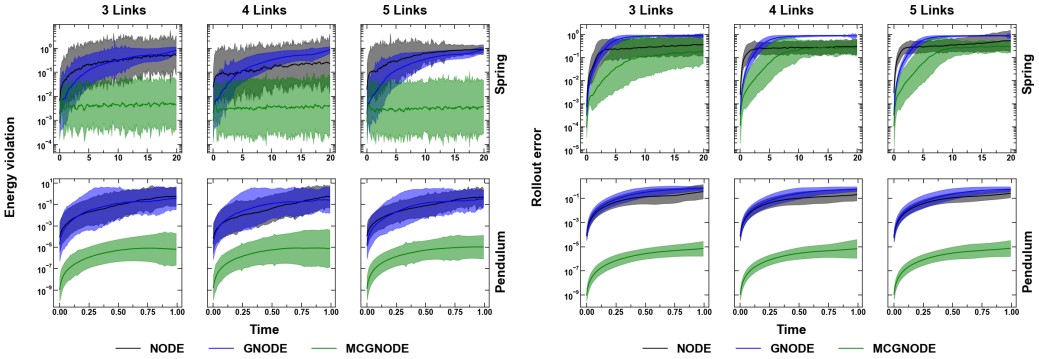

Figure 10: Energy error and trajectory error for NODE, GNODE, and MCGNODE for spring and pendulum systems with 3-,4-, and 5-links. The shaded region represents the 95% confidence interval based on 100 forward simulations.

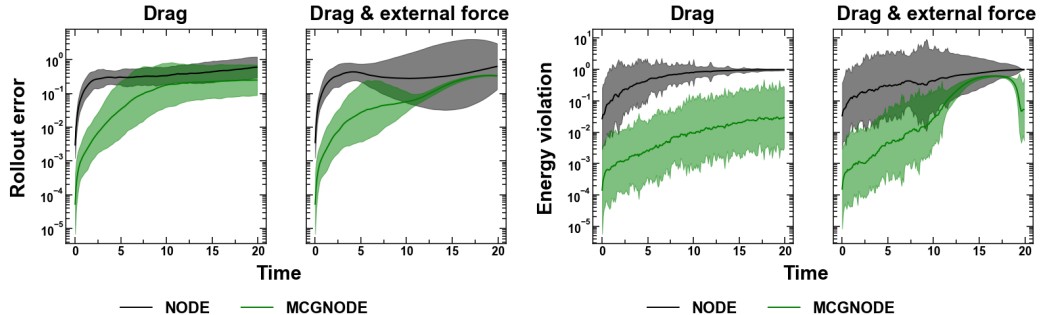

Figure 11: MCGNODE on 5-link spring with non-zero drag and external force with NODE as baseline.

## E  DRAG AND EXTERNAL FORCE

Here, we evaluate the ability of MCGNODE to learn the dynamics of systems subjected to drag linear in velocity (drag force given by $-0.1\dot{q}$) and an external force. Fig. 11 shows the performance of MCGNODE and NODE on a 5-spring system with linear drag and external force. We observe that the MCGNODE exhibits superior performance in both the cases.

## F  TRAINING EFFICIENCY

**Data Efficiency:** Here, we compare the training efficiency of MCGNODE with the NODE and HGN. Despite our best efforts, LGN was unable to learn the dynamics from the small amount of data provided and was exploding up on forward simulation leading to NaN values. Similar behavior was observed on HGN with low values data points. We focus on the best graph-based architecture and compare the performance with the baseline NODE 5-spring and 5-pendulum systems. Figure 12 shows the performance of MCGNODE in comparison to NODE for the dataset sizes of 100, 500, 1000, 5000, and 10000 in terms of the geometric mean of the energy error and rollout error. We observe that the learning of MCGNODE is significantly simplified in comparison to that of NODE— MCGNODE exhibits orders of magnitude better performance in terms of both energy and rollout error for both pendulum and spring systems. Further, we observe that MCGNODE performs orders of magnitude better than HGN.

**Computational Efficiency:** We further compare the computation times of the training and inference phases on GNODE, MCGNODE, HGN and LGN. Tables 1-2 present the results. As visible, MCGNODE, is the fastest on average. The difference with LGN and HGN is the highest in spring systems. MCGNODE benefits from the inductive biases as well as the fact that the neural architecture directly predicts the force (acceleration is computed from this) on each particle. In contrast,

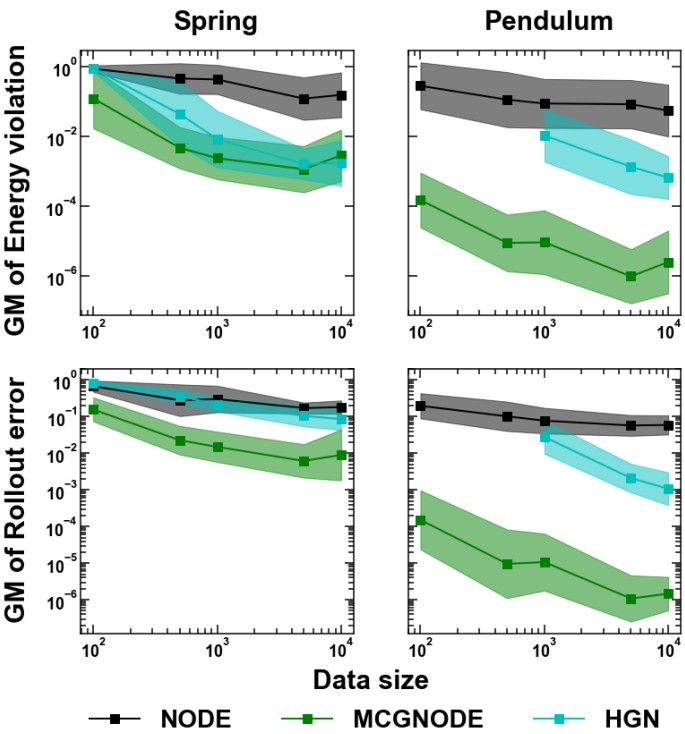

Figure 12: Training efficiency of MCGNODE in comparison with the GNODE. The shaded region represents the 95% confidence interval based on 100 forward simulations.

| Models | Training time (in sec) | Forward Simulation time (in sec) |
|---|---|---|
| MCGNODE | 2433 | 0.87 |
| GNODE | 6341 | 1.02 |
| HGN | 2512 | 0.76 |
| LGN | 55042 | 14.82 |

Table 1: Training and inference times in pendulum systems.

| Models | Training time (in sec) | Forward Simulation time (in sec) |
|---|---|---|
| MCGNODE | 1576 | 0.06 |
| GNODE | 6977 | 0.10 |
| HGN | 14053 | 0.43 |
| LGN | 141962 | 5.81 |

Table 2: Training and inference times in spring systems.

HGN and LGN predict the Hamiltonian and Langrangian, which needs to be further differentiated to obtain acceleration. LGN is the slowest since it requires double differentiation.

## G    ERRORS ON INTERNAL AND EXTERNAL FORCES

In this section, we report the errors in the predicted internal and external forces on each particle of a system. Figs. 13-16 presents the results on GNODE, MCGNODE, LGN and HGN. As visible, the predicted force (represented using dashed lines) overlaps closely with the actual forces (solid line) in MCGNODE (Fig. 14). While the error is also low in GNODE (Fig. 13), it is inferior to MCGNODE showcasing the usefulness of the inductive biases. We also note that the error is the highest in LGN (Fig. 15).

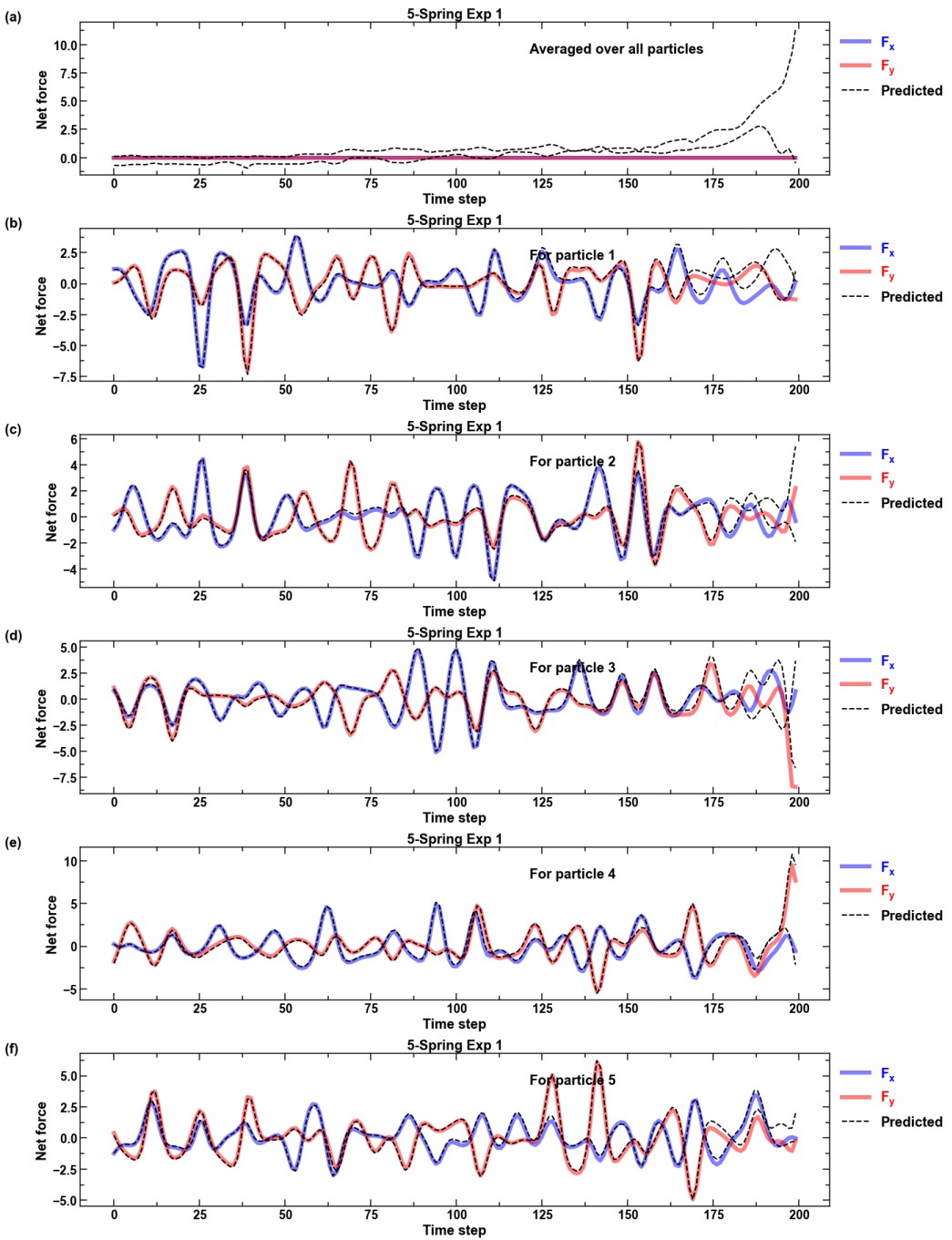

Figure 13: Error in force calculation using GNODE. The solid lines represent the true force and the dashed lines represent the predicted ones. A close overlap among the solid and dashed lines indicate low error. $F_x$ and $F_y$ denote the total forces in the $x$ and $y$ direction respectively. $F_z$ is always 0.

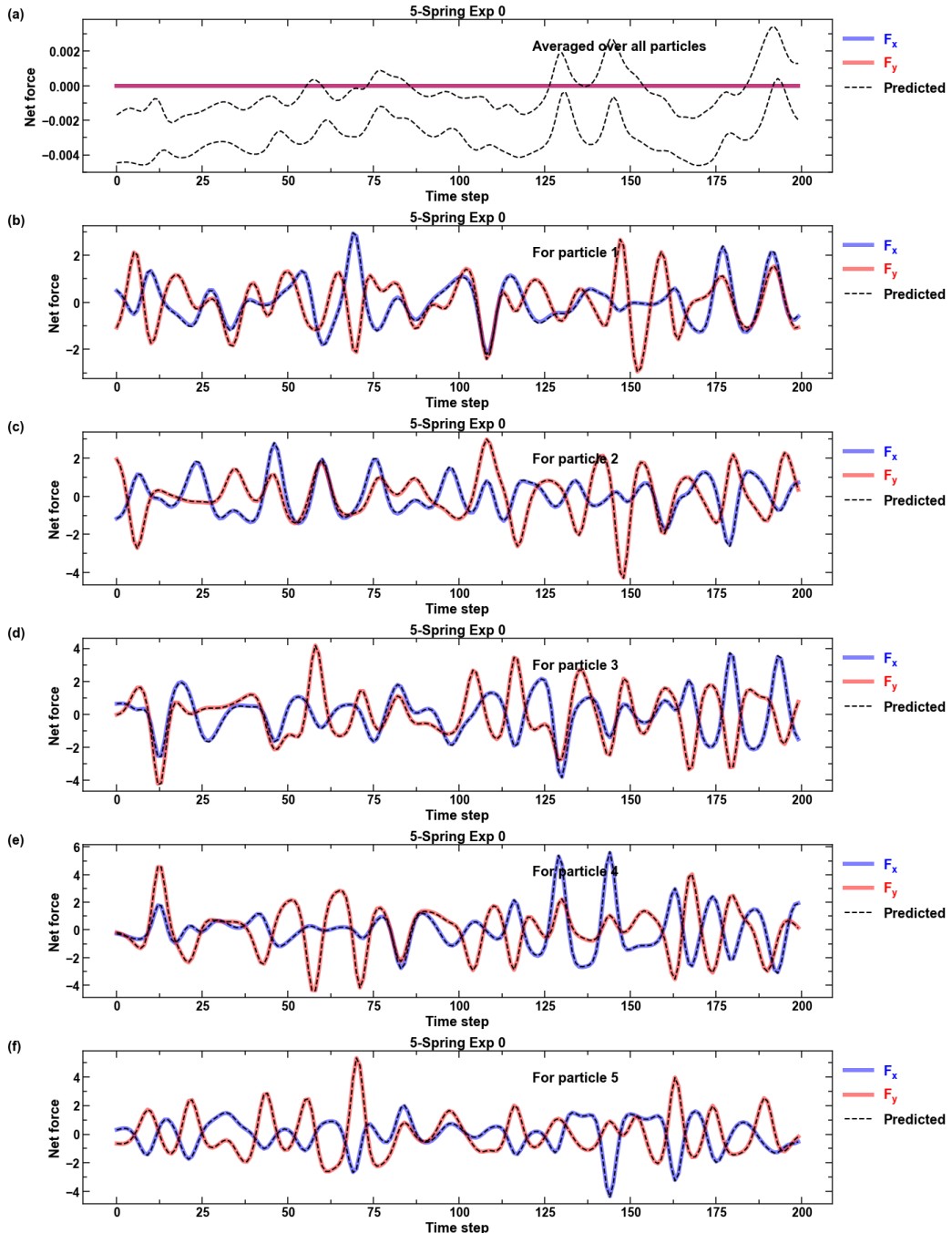

Figure 14: Error in force calculation using MCGNODE. The solid lines represent the true force and the dashed lines represent the predicted ones. A close overlap among the solid and dashed lines indicate low error. $F_x$ and $F_y$ denote the total forces in the $x$ and $y$ direction respectively. $F_z$ is always 0.

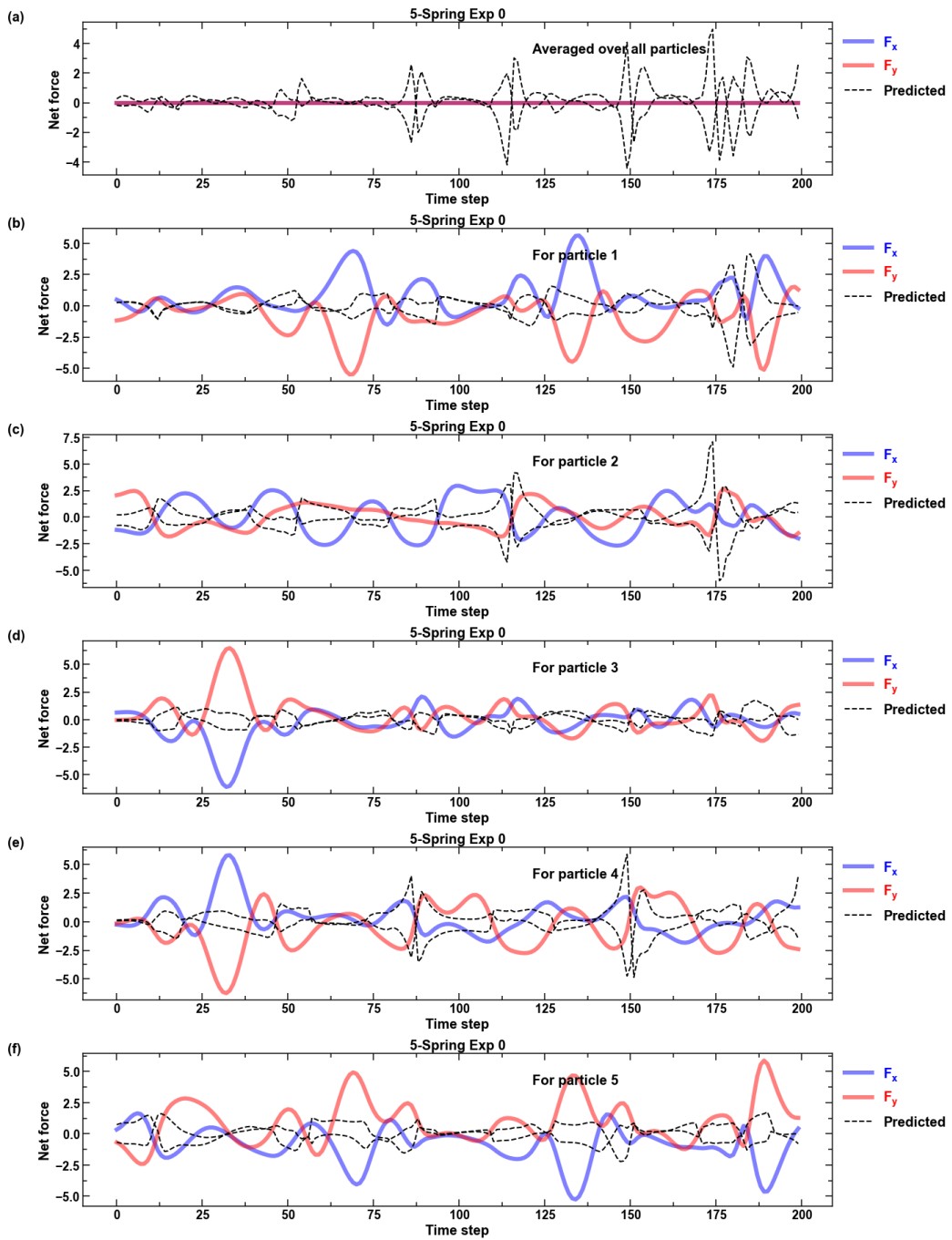

Figure 15: Error in force calculation using LGN. The solid lines represent the true force and the dashed lines represent the predicted ones. A close overlap among the solid and dashed lines indicate low error. $F_x$ and $F_y$ denote the total forces in the $x$ and $y$ direction respectively. $F_z$ is always 0.

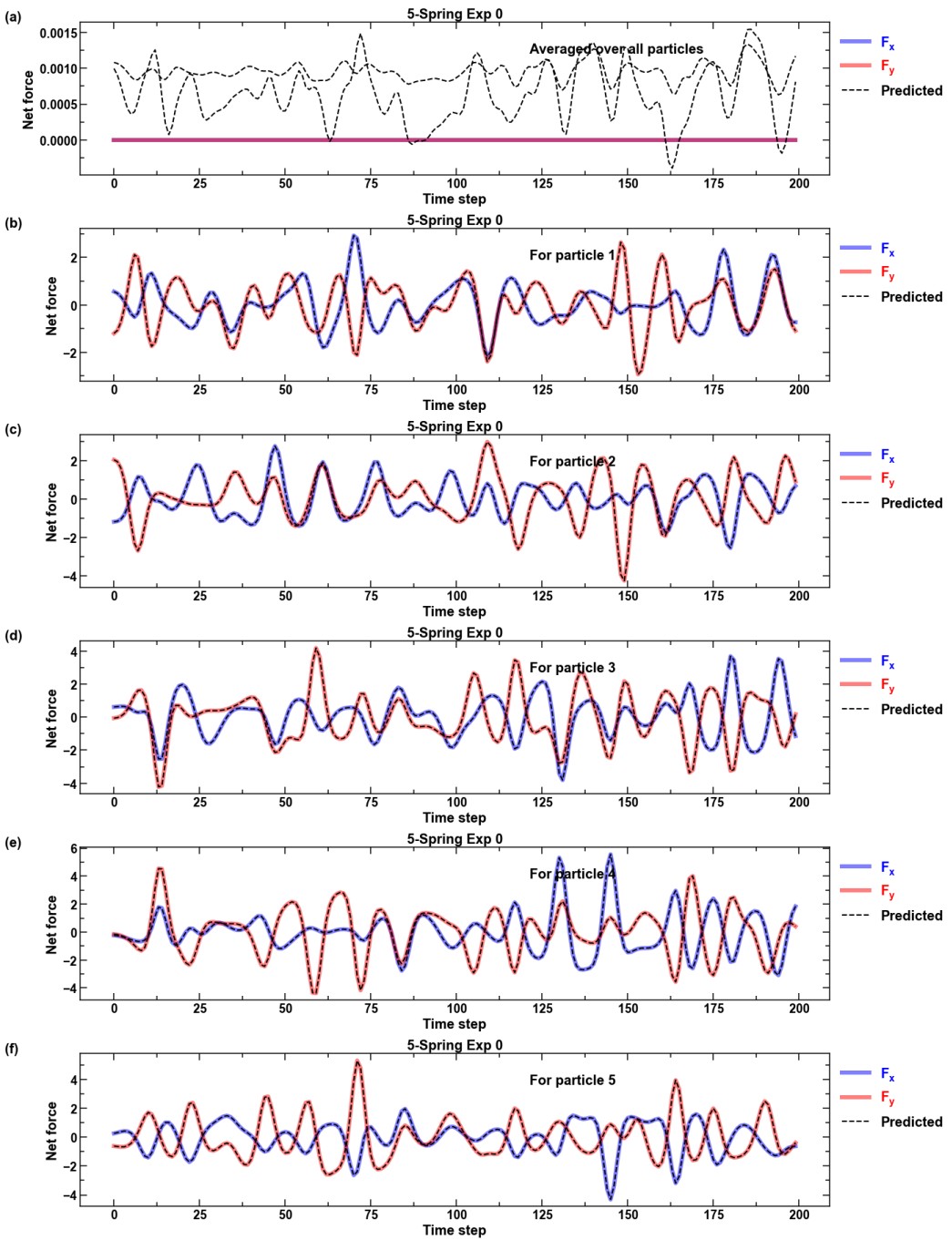

Figure 16: Error in force calculation using HGN. The solid lines represent the true force and the dashed lines represent the predicted ones. A close overlap among the solid and dashed lines indicate low error. $F_x$ and $F_y$ denote the total forces in the $x$ and $y$ direction respectively. $F_z$ is always 0.

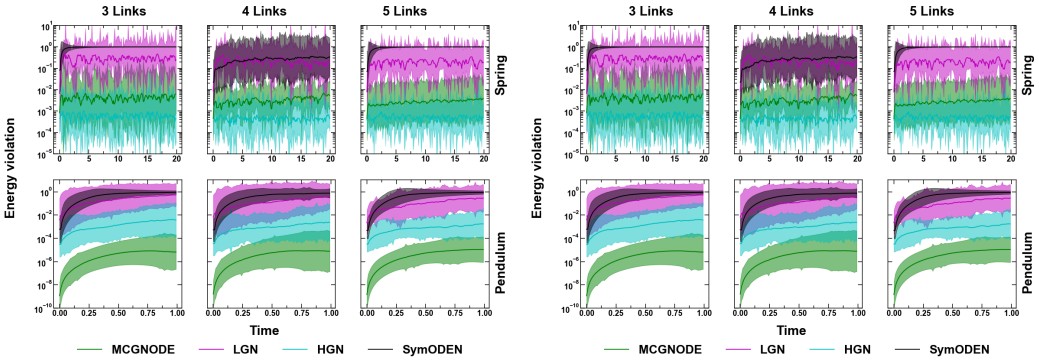

Figure 17: Comparison of MCGNODE, LGN, HGN, and SymODEN. Note that SymODEN is trained and tested for 3-,4-,5-links system, whereas in case of MCGNODE, LGN and HGN training is done only on 5-links system and tested on 3-,4-,5-links.

## H   COMPARISON OF MCGNODE, LGN, HGN, SYMODEN

Fig. 17 shows the performance of an additional baseline, namely, SymODE-Net (SymODEN) Zhong et al. (2019) on 3,4,5-spring and pendulum systems. SymODE-Net being a feed-forward MLP is transductive in nature and thus cannot do zero-shot knowledge transfer. Hence, it is trained and tested on each system separately. We observe that the performance of SymODE-Net is comparable to LGN but poorer than HGN and MCGNODE.

## I   EXPERIMENTAL SYSTEMS

### I.1   SIMULATION ENVIRONMENT

All the simulations and training were carried out in the JAX environment (Schoenholz & Cubuk, 2020; Bradbury et al., 2020). The graph architecture was developed using the jraph package (Godwin* et al., 2020). The experiments were conducted on a machine with Apple M1 chip having 8GB RAM and running MacOS Monterey.

### I.2   $n$-PENDULUM

In an $n$-pendulum system, $n$-point masses, representing the bobs, are connected by rigid bars which are not deformable. These bars, thus, impose a distance constraint between two point masses as

$$||q_i - q_{i-1}|| = l_i \tag{15}$$

where, $l_i$ represents the length of the bar connecting the $(i-1)^{th}$ and $i^{th}$ mass. This constraint can be differentiated to write in the form of a Pfaffian constraint as

$$q_i \dot{q}_i + q_{i-1} \dot{q}_{i-1} = 0 \tag{16}$$

Note that such constraint can be obtained for each of the $n$ masses considered to obtain the $A(q)$.

The mass of each of the particles is $m_i$ and the force on each bob due to the acceleration due to gravity in $m_i g$. This force will be compensated by the constraint forces, which in turn governs the motion of the pendulum. For LGN, the Lagrangian of this system has contributions from potential energy due to gravity and kinetic energy. Thus, the Lagrangian can be written as

$$\mathcal{L} = \sum_{i=1}^{n} \left( 1/2 m_i \dot{q}_i^{\top} \dot{q}_i - m_i g y_i \right) \tag{17}$$

And, for HGN, the Hamiltonian is given by

$$\mathcal{H} = \sum_{i=1}^{n} \left( 1/2 m_i \dot{q}_i^{\top} \dot{q}_i + m_i g y_i \right) \tag{18}$$

where $g$ represents the acceleration due to gravity in the $y$ direction.

### I.3 $n$-SPRING SYSTEM

In this system, $n$-point masses are connected by elastic springs that deform linearly with extension or compression. Note that similar to a pendulum setup, each mass $m_i$ is connected only to two masses $m_{i-1}$ and $m_{i+1}$ through springs so that all the masses form a closed connection. Thus, the force experienced by each bob $i$ due to a spring connecting $i$ to its neighbor $j$ is $f_{ij} = 1/2k(||q_i - q_j|| - r_i)$. For LGN, the Lagrangian of this system is given by

$$\mathcal{L} = \sum_{i=1}^{n} 1/2 m_i \dot{q}_i^\top \dot{q}_i - \sum_{i=1}^{n} 1/2 k(||q_{i-1} - q_i|| - r_0)^2 \tag{19}$$

And, for HGN, the Hamiltonian by:

$$\mathcal{H} = \sum_{i=1}^{n} 1/2 m_i \dot{q}_i^\top \dot{q}_i + \sum_{i=1}^{n} 1/2 k(||q_{i-1} - q_i|| - r_0)^2 \tag{20}$$

where $r_0$ and $k$ represent the undeformed length and the stiffness, respectively, of the spring.

## J IMPLEMENTATION DETAILS

### J.1 DATASET GENERATION

**Software packages:** numpy-1.22.1, jax-0.3.0, jax-md-0.1.20, jaxlib-0.3.0, jraph-0.0.2.dev
**Hardware:** Chip: Apple M1, Total Number of Cores: 8 (4 performance and 4 efficiency), Memory: 8 GB, System Firmware Version: 7459.101.3, OS Loader Version: 7459.101.3

For NODE, all versions of GNODE, LGN: All the datasets are generated using the known Lagrangian of the pendulum and spring systems, along with the constraints, as described in Section I. For each system, we create the training data by performing forward simulations with 100 random initial conditions. For the pendulum system, a timestep of $10^{-5}s$ is used to integrate the equations of motion, while for the spring system, a timestep of $10^{-3}s$ is used. The velocity-Verlet algorithm is used to integrate the equations of motion due to its ability to conserve the energy in long trajectory integration.

While for HGN, datasets are generated using Hamiltonian mechanics. Runge-Kutta integrator is used to integrate the equations of motion due to the first order nature of the Hamiltonian equations in contrast to the second order nature of LGN, NODE, and all versions of GNODE.

From the 100 simulations for pendulum and spring system, obtained from the rollout starting from 100 random initial conditions, 100 data points are extracted per simulation, resulting in a total of 10000 data points. Data points were collected every 1000 and 100 timesteps for the pendulum and spring systems, respectively. Thus, each training trajectory of the spring and pendulum systems are $10s$ and $1s$ long, respectively. Here, we do not train from the trajectory. Rather, we randomly sample different states from the training set to predict the acceleration.

### J.2 ARCHITECTURE

For NODE, we use a fully connected feed-forward neural network with 2 hidden layers, each having 16 hidden units with a square-plus activation function. For all versions of GNODE, graph architecture is used with each MLP consisting of two layers with 5 hidden units. One round of message passing is performed for both pendulum and spring. For LGN and HGN, full graph architecture is used with each MLPs consist of two layers with 16 hidden units and one round of message passing were performed. The hyper-parameters were optimized based on the good practices of the model development.

### J.3 EDGE FEATURES: POSITION VS VELOCITY

Figure 18 shows the loss curve with relative position and relative velocity as edge features. We observe that when relative velocity is provided as the edge feature, model struggles to learn and saturates quickly.

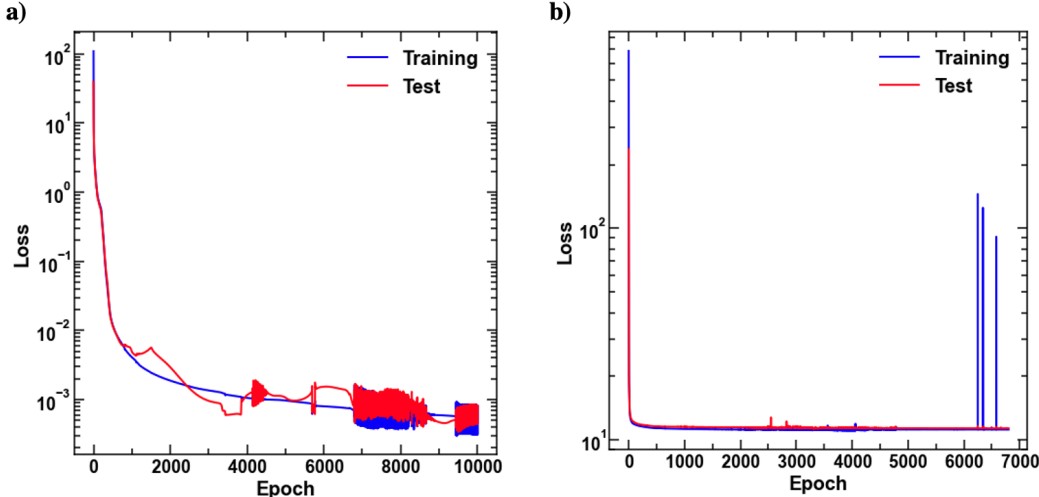

Figure 18: Loss curve when (a) relative position is used as edge feature and (b) relative velocity is used as edge feature.

## J.4 TRAINING DETAILS

The training dataset is divided in 75:25 ratio randomly, where the 75% is used for training and 25% is used as the validation set. Further, the trained models are tested on its ability to predict the correct trajectory, a task it was not trained on. Specifically, the pendulum systems are tested for $1s$, that is $10^5$ timesteps, and spring systems for $20s$, that is $2 \times 10^4$ timesteps on 100 different trajectories created from random initial conditions. All models are trained for 10000 epochs. A learning rate of $10^{-3}$ was used with the Adam optimizer for the training.

•**NODE**

| Parameter | Value |
|---|---|
| Hidden layer neurons (MLP) | 16 |
| Number of hidden layers (MLP) | 2 |
| Activation function | squareplus |
| Optimizer | ADAM |
| Learning rate | $1.0e^{-3}$ |
| Batch size | 100 |

•**GNODE, CGNODE, CDGNODE, MCGNODE**

| Parameter | Value |
|---|---|
| Node embedding dimension | 5 |
| Edge embedding dimension | 5 |
| Hidden layer neurons (MLP) | 5 |
| Number of hidden layers (MLP) | 2 |
| Activation function | squareplus |
| Number of layers of message passing | 1 |
| Optimizer | ADAM |
| Learning rate | $1.0e^{-3}$ |
| Batch size | 100 |

•**LGN, HGN**

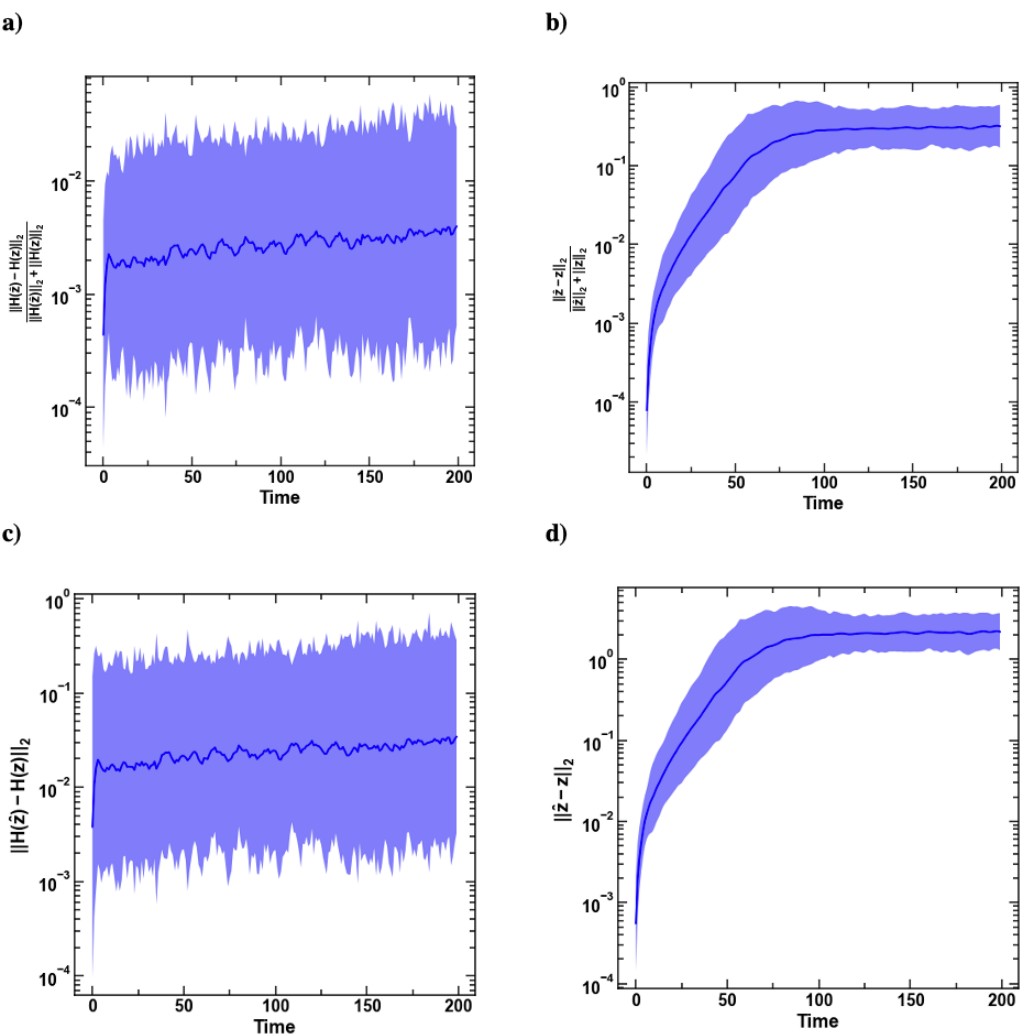

Figure 19: Plot comparing (a) relative energy error, (b) relative trajectory error, (c) absolute energy error, (d) absolute trajectory error for the 5-Spring-MCGNODE system

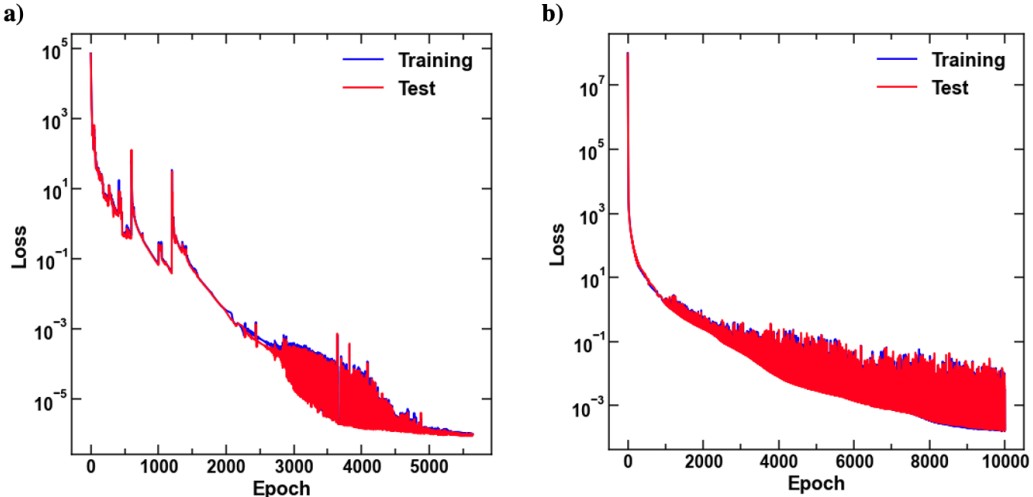

Figure 20: Loss curve for peridynamics system using (a) MCGNODE architecture and (b) GNODE architecture.

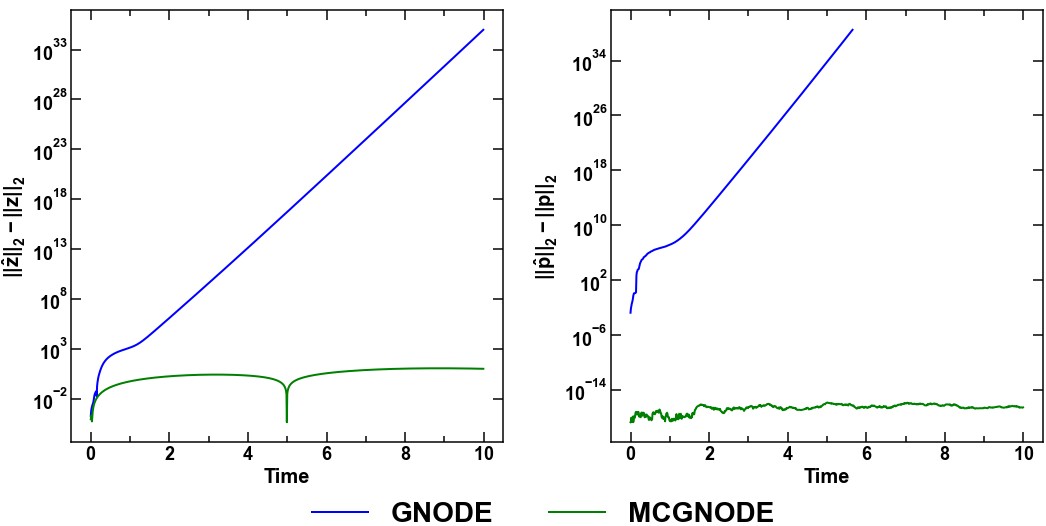

Figure 21: Absolute trajectory error error and absolute momentum error for peridynamics simulation using of GNODE, and MCGNODE.

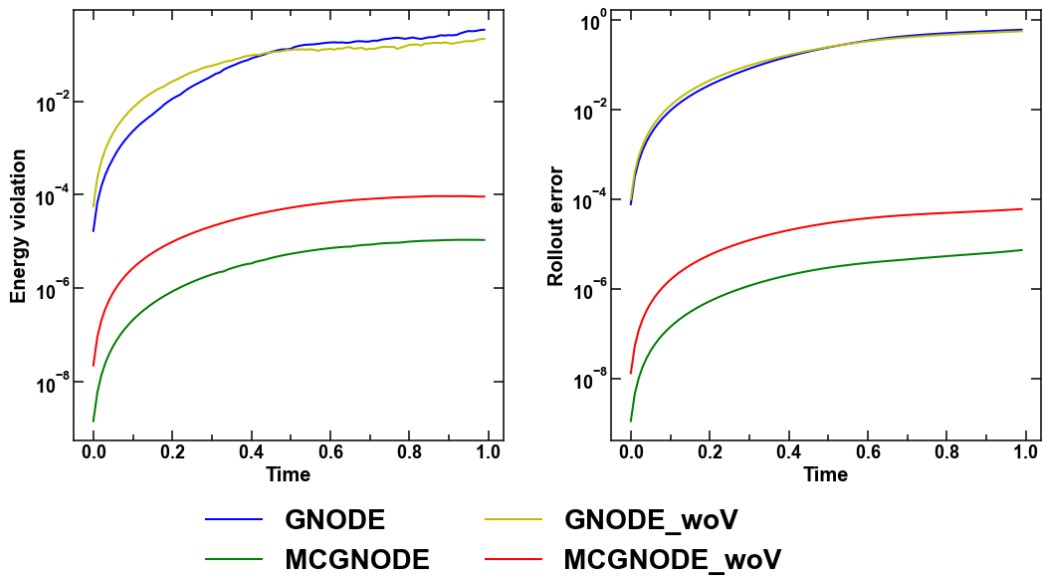

Figure 22: Comparison of Energy violation and rollout error for 5-link pendulum using GNODE, MCGNODE, GNODE-woV (GNODE without velocity as input) and MCGNODE-woV (MCGNODE without velocity as input).

| Parameter | Value |
|---|---|
| Node embedding dimension | 8 |
| Edge embedding dimension | 8 |
| Hidden layer neurons (MLP) | 16 |
| Number of hidden layers (MLP) | 2 |
| Activation function | squareplus |
| Number of layers of message passing | 1 |
| Optimizer | ADAM |
| Learning rate | $1.0e^{-3}$ |
| Batch size | 100 |

## K   GRAPH ARCHITECTURES: GNODE, CGNODE, CDGNODE, MCGNODE

The detailed differences between each of the architectures presented in the work are as follows.

• **GNODE**
**Summary**: Graph version of NODE that allows inductivity to unseen system sizes.
**Node inputs**: position ($q$), velocity ($\dot{q}$), and type ($t$)
**Edge inputs**: $w_{ij} = (q_i - q_j)$
**Node output**: acceleration of each particle ($\ddot{q}_i$). Note that we do not use Eq. 5 here to predict $\dot{q}$. Instead, we directly predict the acceleration using $\ddot{q}_i = \hat{F}(q, \dot{q}, t)$, where $\hat{F}$ is the approximate dynamics learned by the GNODE. Here, predicting force and acceleration are equivalent problems since the acceleration is a simple scaling of the force when the masses of the particles are equal as in the present case.
**Edge output**: Nil.
**Architecture**: Fig. 8. Here, GNODE directly predicts the acceleration and not the force although they are equivalent problems.
**Loss function**: $\mathcal{L} = \frac{1}{n} \left( \sum_{i=1}^{n} \sum_{t=2}^{\mathcal{T}} \left( \ddot{q}_i^{\mathbb{T},t} - \hat{\ddot{q}}_i^{\mathbb{T},t} \right)^2 \right)$

**Assumed knowns**: None, since we do not use Eq. 5 here.
**Inductive bias**: Modeling the physical system as a graph with the particles as nodes and connections

between them as edges.

### •CGNODE
**Summary**: Same as GNODE except for the property that the neural network predicts the force from which acceleration is computed using Eq. 5.
**Node inputs**: Same as in GNODE
**Node output**: The force $N_i$ on each particle $i$
**Edge output**: Nil, same as in GNODE
**Architecture**: Fig. 8
**Loss function**: Same as in GNODE. The acceleration is obtained using Eq. 5, where $N$ is obtained from the CGNODE, $A(q)$ is assumed to be known, $M$ is the learnable diagonal mass matrix, $\Pi$ is the known external force, $C$ is 0, $\Upsilon$ is the learnable external drag. Note that unless specified, $\Pi$ and $\Upsilon$ is considered to be 0, that is, the system is not subjected to external force or drag. Details below.
**Assumed knowns**: (i) Governing equation for acceleration as given by Eq. 5. (ii) $k$ constraints (holonomic or Pfaffian) in the physical system represented by $A(q) \in \mathbb{R}^{kD}$ based on the topology and nature (rigid/deformable)) of the system. For instance, the bar in the pendulum is rigid whereas the springs are deformable.
**Inductive bias**: Same as GNODE + explicit constraints as given by $A(q)$ and acceleration as given by Eq. 5.

### •CDGNODE
**Summary**: Same as CGNODE except that the architecture is modified to decouple the computations of internal and body forces. The raw node and edge inputs are the same. However, since we learn two representations per node, global and local, these inputs are used in the following manner:
**Input for global node representations**: position ($q$) and velocity ($\dot{q}$); these features are not involved in message passing.
**Input for local node representations**: type ($t$).
**Input for local edge representations**: $w_{ij} = (q_i - q_j)$, i.e., same as in GNODE. Note that node local input and edge inputs are involved in message passing.
**Node output**: Same as in CGNODE, but the local node output and the global node output are concatenated and passed through an MLP to obtain the total node force ($N_i$) as shown in Fig. 1. The intuition is that the local node output represents the internal forces (for instance, the spring forces) and the global node output represents the body forces (for instance, the gravitational force), which when combined gives the total force on each particle. This design thus enables more expressive power for the neural network.
**Edge outputs**: Nil, same as in CGNODE.
**Architecture**: Fig. 1.
**Loss function**: Same as in CGNODE.
**Assumed knowns**: Same as in CGNODE.
**Inductive bias**: Same as in CGNODE + decoupling the computation of internal forces (which depend on the topology and is a function of the $w_{ij}$) and body forces (which typically depends on the position of the particle rather than the topology of the system, for instance, gravitational or electromagnetic force).

### •MCGNODE
**Summary**: Same as CDGNODE except that the architecture is modifed to enforce Newton's third law by incorporation of an additional MLP to predict the individual interaction forces at the edge level.
**Node global input, node local input, edge input**: Same as in CDGNODE.
**Edge output**: Force $f_{ij}$ on particle $i$ due to the interaction from the neighboring particle $j$.
**Node output**: Total force $N_i$ for each particle given by $N_i = N_{gi} + \sum_{\mathcal{N}_i} f_{ij}$, where $\mathcal{N}_i$ represents of the set of all the $j$ neighbors of $i$ and $N_{gi}$ represents the force due to external fields obtained by passing the global node output through an MLP.
**Architecture**: Fig. 8 with a difference that there are two MLPs, one to compute $N_{gi}$ which takes as input the global node representation and a second that takes the edge representation as input to predict the $f_{ij}$. We will include a separate figure to explcitly differentiate between CDGNODE and MCGNODE.

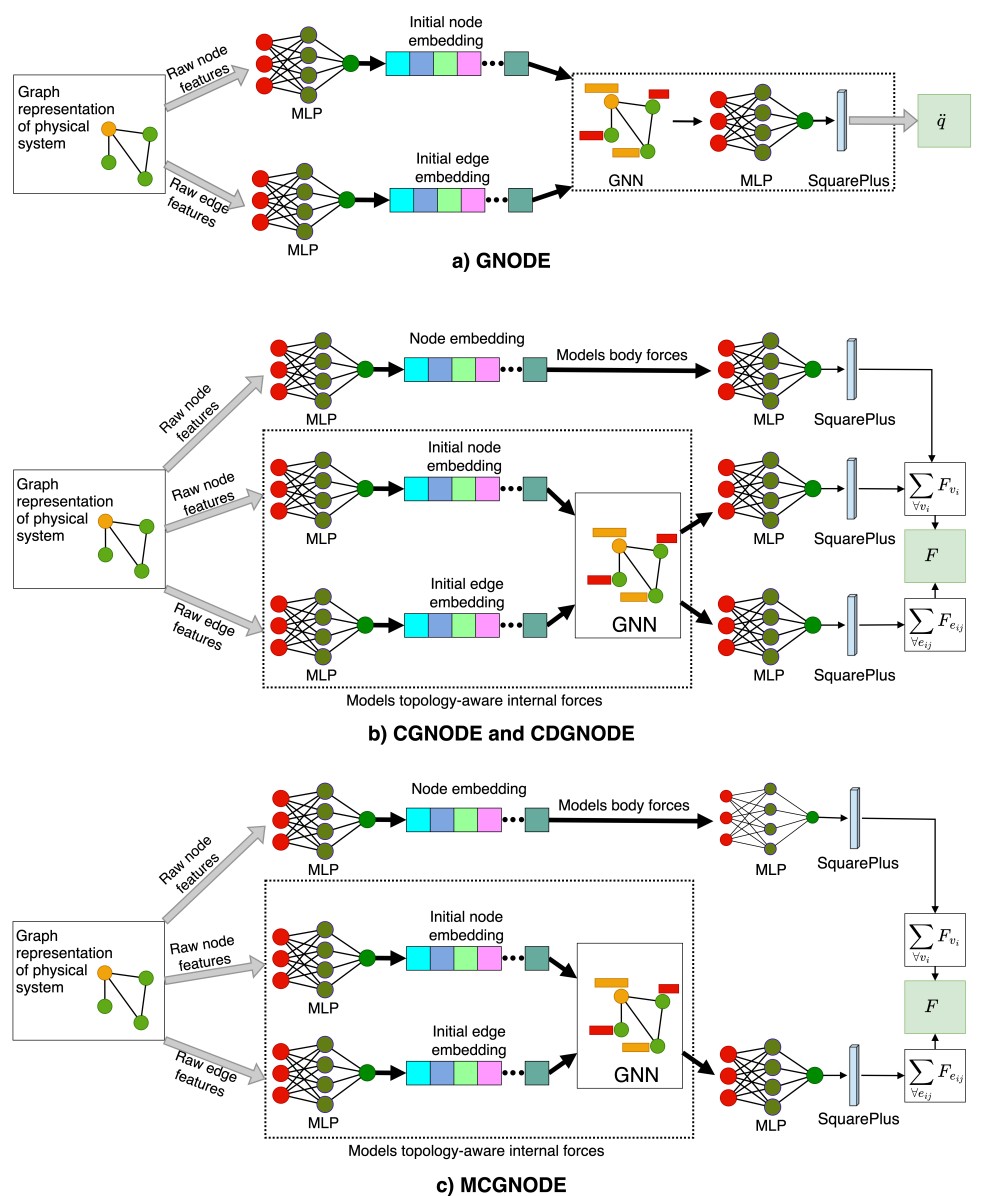

Figure 23: Architecture of (a) GNODE, (b) CGNODE, CDGNODE and (c) MCGNODE. Note that the architecture of CGNODE and CDGNODE are equivalent.

**Loss function**: Same as in CGNODE.
**Assumed knowns**: same as in CGNODE.
**Inductive bias**: Same as in CDGNODE + Newton's third law, that is, the internal forces between the particles are equal and opposite.

## L  MCGNODE VS GNS

The tables 3, 4 demonstrate that GNS performs inferior to MCGNode.

| Time(s) | GNS | GNODE | MCGNODE |
|---------|------|-------|---------|
| 0.00 | 0.74 | 0.92 | 0.26 |
| 1.00 | 0.88 | 1.00 | 0.28 |
| 2.00 | 0.90 | 1.00 | 0.30 |
| 3.00 | 0.89 | 1.00 | 0.36 |
| 4.00 | 0.81 | 1.00 | 0.52 |
| 5.00 | 0.84 | 1.00 | 0.80 |
| 6.00 | 0.88 | 1.00 | 0.81 |
| 8.00 | 0.88 | 1.00 | 0.76 |
| 9.00 | 0.85 | 1.00 | 0.75 |
| 9.90 | 0.84 | 1.00 | 0.70 |

Table 3: Momentum error

| Time(s) | GNS | GNODE | MCGNODE |
|---------|------|-------|---------|
| 0.00 | 0.00 | 0.00 | 0.00 |
| 1.00 | 1.00 | 0.96 | 0.01 |
| 2.00 | 1.00 | 1.00 | 0.02 |
| 3.00 | 1.00 | 1.00 | 0.04 |
| 4.00 | 1.00 | 1.00 | 0.06 |
| 5.00 | 1.00 | 1.00 | 0.06 |
| 6.00 | 1.00 | 1.00 | 0.09 |
| 7.00 | 1.00 | 1.00 | 0.12 |
| 8.00 | 1.00 | 1.00 | 0.16 |
| 9.90 | 1.00 | 1.00 | 0.21 |

Table 4: Rollout error

