# OpenReview forum: "Enhancing the Inductive Biases of Graph Neural ODE for Modeling Physical Systems"
_ICLR.cc/2023/Conference — ICLR 2023 poster_

### Official Review · Reviewer_zGuL · 2022-10-16

**Confidence:** 3
**Correctness:** 4
**Technical Novelty And Significance:** 3
**Empirical Novelty And Significance:** 2
**Recommendation:** 8

**Clarity, Quality, Novelty And Reproducibility:**

The paper is well written and clear. The novelty is sufficient as their way of introducing the physics-based inductive biases to the ODE models are different than previous work.

**Strength And Weaknesses:**

Strength:

- The authors have done a good job in bridging the physics and ML by bringing such inductive biases which really help learn the dynamics in these systems effectively.

Weakness:

- These inductive biases work well in physics-based systems such as pendulum and spring. The authors have not investigated its efficacy in more general dynamical systems such as time-series.

- Related to the previous point, I think the examples in this paper are not diverse and only show the physical systems, pendulum and spring.

**Summary Of The Paper:**

Neural ODEs are powerful models for learning dynamical systems, however, they cannot be generalized the systems with larger sizes when trained on the smaller systems. In order to solve this problem this paper proposes a graph-based neural ODE (GNODE) which can be inductive and generalize to larger systems with the similar dynamics such as pendulum and springs. Base on success from adding physics-based inductive biases in previous works such as Lagrangian neural networks (LNN) and Hamiltonian neural networks (HNN), this paper also explores the effectiveness of adding various types of physics-based inductive biases to the GNODEs. These inductive biases are (1) Decoupling the dynamics and constraints (CGNODE), (1) Decoupling the internal and body forces (CDGNODE), and (3) Newton’s third law (MCGNODE). They evaluate the models based on three metrics: rollout error, energy violation error, and momentum conservation error. The results show the performance of GNODE model, its different versions with distinct inductive biases, and two competing methods (LGN and HGN). Overall, the performance of MCGNODE is the best among GNODE models in terms of those three evaluation metrics, and it also outperforms the competing methods in the pendulum and spring systems.

**Summary Of The Review:**

I liked the idea of introducing three different types of physics-based inductive biases to the graph-based ODE models. It is always helpful to add known governing rules of the systems to the models in order to guide them to learn the true dynamics. However, I have the following concerns/questions:

- This is related to the usefulness of this method in other dynamical systems. I am convinced that this method should work great in the physical systems where we exactly know some of the underlying rules. However, can it still work so well in other systems such as time-series data? In other words, can bringing such inductive biases still help in the systems that we do not know for sure such assumptions do not hold?

- I think it would really help the reach of paper if it includes time-series data, as right now, the diversity of examples are limited to physical systems.

Some minor comments:

- Typos:
           an node  --> a node
           the learn --> to learn

- Should eq (11) have another summation over t?
- Why the authors use Verlet algorithm to compute the accelerations in eq (12)? Shouldn't the acceleration values be observed in such systems? How to choose \delta t and how accurate are the estimation of the accelerations using (12)? It would be great if the authors clarify this.

---

> ### Author Response · Authors · 2022-11-15
> **Response to Reviewer zGuL**
>
> **These inductive biases work well in physics-based systems such as pendulum and spring. The authors have not investigated its efficacy in more general dynamical systems such as time-series. Related to the previous point, I think the examples in this paper are not diverse and only show the physical systems, pendulum and spring.**
>
> *Response:* The focus of the present work is to enable accelerated modeling of physical systems by learning the dynamics directly from the trajectory. There is a growing literature of learning the dynamics of physical systems using machine learning, which has direct application in areas such as robotics and control, computer graphics, and even in scientific domains such as physics, biology, astronomy, and engineering. In the present work, we show that GNNs with appropriate inductive biases can learn the dynamics from very small amount of data and extrapolate to much larger system sizes. We also critically analyze the role of each of the inductive biases in enhancing the performance. Indeed, the approach mentioned here can be extrapolated to other dynamics systems such as time-series. However, such extension is non-trivial, system specific, and requires further work.
>
> To address this comment, we propose to modify the title of the paper to *"ENHANCING THE INDUCTIVE BIASES OF GRAPH NEURAL ODE FOR MODELING **PHYSICAL** SYSTEMS"*. In addition, we have added an additional comment in Sec. 5 on the potential future work towards extending the approach to other dynamical systems.
>
> **I liked the idea of introducing three different types of physics-based inductive biases to the graph-based ODE models. It is always helpful to add known governing rules of the systems to the models in order to guide them to learn the true dynamics. However, I have the following concerns/questions:**
>
> **This is related to the usefulness of this method in other dynamical systems. I am convinced that this method should work great in the physical systems where we exactly know some of the underlying rules. However, can it still work so well in other systems such as time-series data? In other words, can bringing such inductive biases still help in the systems that we do not know for sure such assumptions do not hold?**
>
> *Response:* Indeed the approach mentioned here can potentially be applied to other time-series data for which the underlying governing equations or conservation laws are known. However, this requires additional investigation which is beyond the scope of the present work. To address this comment, we have added an additional comment in Sec. 5 on the potential future work towards extending the approach to other dynamical systems. In addition, as mentioned above, we propose to change the title to not leave any room for ambiguity that the present work focuses on physical systems.
>
> **I think it would really help the reach of paper if it includes time-series data, as right now, the diversity of examples are limited to physical systems.**
>
> *Response:* We thank the reviewer for this suggestion. As mentioned earlier, we have considered three varieties of physical systems, namely $n$-pendulum, $n$-spring, and a $3D$ deformable body. Further, we have now included additional systems with linear drag and external forces in Appendix (App. E, Fig. 11). We will indeed consider extending the work to other dynamical systems as part of our future work.
>
> **Some minor comments:**
>
> **Typos: an node --> a node, the learn --> to learn**
>
> *Response:* Thank you. Both typos have now been corrected.
>
> **Should eq (11) have another summation over t?**
>
> *Response:* Thanks for pointing this out. Eq. (11) has now been corrected.
>
> **Why the authors use Verlet algorithm to compute the accelerations in eq (12)? Shouldn't the acceleration values be observed in such systems? How to choose \delta t and how accurate are the estimation of the accelerations using (12)? It would be great if the authors clarify this.**
>
> *Response:* Note that the canonical coordinates employed in Newtonian mechanics are the position ($q$) and velocity ($\dot{q}$) (or momentum) as these quantities are the direct observables. Verlet algorithm is obtained by subtracting the Taylor series expansion of $r(t+\Delta t)$ with $r(t-\Delta t)$. The expression thus obtained is time-reversible and has error to the order of ($\mathcal{O}(\Delta t^4)$). Due to the time-reversible nature, Verlet algorithm is energy conserving (symplectic), and hence performs better than other traditional integrators such as RK4 in predicting the long-term trajectory of dynamical systems without drift.
>
> We have added the above discussion to the Appendix (App. A) with a reference from Sec 2 in the main paper.

---

> > ### Comment · Reviewer_zGuL · 2022-11-23
> > **Thanks for your response**
> >
> > My concerns are addressed and the quality of the paper has increased upon answering the reviewers' comments. So, I will increase my score.

---

> > > ### Author Response · Authors · 2022-11-24
> > > **Thank you!**
> > >
> > > We thank and appreciate the reviewer for engaging in the discussion, positive comments, and raising the score!
> > >
> > > Thank you,
> > >
> > > Authors

---

> ### Author Response · Authors · 2022-11-17
> **Looking forward to feedback from Reviewer zGuL**
>
> Dear Reviewer,
>
> Thank you for taking time for the review and positive feedback. As outlined below, we have addressed all the concerns raised by the reviewer. Specifically, we have now **added new text** to improve the clarity regarding the concerns and have **modified the title** of the paper. If there are any outstanding concerns, please do let us know. Otherwise, we would appreciate if you can raise the score of the paper.
>
> Looking forward to your response.
>
> Thank you,
>
> Authors

---

### Official Review · Reviewer_okft · 2022-10-22

**Confidence:** 3
**Correctness:** 3
**Technical Novelty And Significance:** 3
**Empirical Novelty And Significance:** 2
**Recommendation:** 6

**Clarity, Quality, Novelty And Reproducibility:**

- This manuscript is readable.
- I think the proposed method contains novelty.
- There are some concerns about the evaluation. It is necessary to examine whether the comparison method is sufficient.

**Strength And Weaknesses:**

S1. A new graph-based NODE is proposed, which can embed several physics-informed inductive biases.
S2. In experiments, it is shown that the proposed method can predict trajectories more accurately than NODE while successfully capturing the energy conservation law.
S3. It is interesting that predictions can be made for unseen systems.

W1. Comparison methods seem to be lacking. For example, in Figure 2, it would be good to have a comparison not only with NODE, but also with models that can use physical knowledge (e.g. Symplectic ODE-Net [R1]).
W2. The task of predicting unseen systems is very interesting, but I think other approaches (e.g. meta-learning ) are working on similar tasks. I think it would be good to add a discussion of this in the related research section.
W3. There are several unclear points as follows:
  - This study is formulated based on the Lagrangian form. What is the reason for choosing the Lagrangian form? Also, are the ideas in this study applicable to Hamiltonian form?
  - Can you also briefly tell me about the advantages of studying physical systems based on GNN?
  - The more inductive biases you consider, the more robust you will be against sparsity and noise in your data. If you have any views or experimental results, please let me know.

[R1] Zhong et. al., Symplectic ODE-Net: Learning Hamiltonian Dynamics with Control, ICLR, 2020.
[R2] Lee et. al., IDENTIFYING PHYSICAL LAW OF HAMILTONIAN SYSTEMS VIA META-LEARNING, ICLR, 2021.

**Summary Of The Paper:**

This paper discusses on inductive biases for effective learning of physical dynamics based on a data-driven approach. The proposed method, GNODE, is an extension of neural ODE and can describe physical dynamics using graph structures and can embed some physics-informed inductive biases. In the experiments, the effectiveness of the proposed method is verified by comparing it with NODE and graph-based NN models (LGN and HGN). It is also applied to transductive settings to achieve good performance.

**Summary Of The Review:**

Overall, it addresses an interesting problem and the method is novel. However, there are concerns about the choice of comparison methods and some ambiguities in the manuscript.

---

> ### Author Response · Authors · 2022-11-15
> **Response to Reviewer okft**
>
> **W1. Comparison methods seem to be lacking. For example, in Figure 2, it would be good to have a comparison not only with NODE, but also with models that can use physical knowledge (e.g. Symplectic ODE-Net [R1]).**
>
> **[R1] Zhong et. al., Symplectic ODE-Net: Learning Hamiltonian Dynamics with Control, ICLR, 2020.**
>
> *Response:* Symplectic ODE-Net suggested by the reviewer employs a decoupled Hamiltonian formulation where the potential energy is learned using a neural network and the mass is parametrized using learnable weights. This is an additional inductive bias employed on HNN, exploiting the structure of the Hamiltonian. Indeed, this is similar to the decoupled GNode employed in the present work. In fact, all the three inductive biases that we have employed to GNode can be effectively applied to LGN and HGN as well. This can potentially improve the performance of these models. However, in the present work, we focussed on GNode due to its ability to directly predict the acceleration, thereby reducing the computational and training time on complex systems. Thus, the core contribution of the present work is the inductive biases themselves, which can enhance the performance of GNNs for modeling physical systems. It is also worth noting that the SymODEN is based on a feed-forward MLP and hence is not inductive as in the GNNs. Thus, the training and inference can be performed on the same system size and not on a different system.
>
> To address this comment, we have included the new baseline SymODE-Net (SymODEN) in the Appendix for the 3,4,5-spring and pendulum systems (see App. H, Fig. 17). SymODE-Net being a feed-forward MLP is trained and tested on each system separately. We observe that the performance of SymODE-Net is comparable to LGN but poorer than HGN and MCGNode.
>
> **W2. The task of predicting unseen systems is very interesting, but I think other approaches (e.g. meta-learning) are working on similar tasks. I think it would be good to add a discussion of this in the related research section.**
>
> **[R2] Lee et. al., IDENTIFYING PHYSICAL LAW OF HAMILTONIAN SYSTEMS VIA META-LEARNING, ICLR, 2021.**
>
> *Response:*  we have added the below discussion to Sec 1.
>
> > In this context, there exists work on *few-shot* generalizability to unseen physical systems  through *meta-learning* [R2]. Meta-learning assumes the availability of limited amount of training data to adapt to an unseen system (task). This is different from our objective where we perform zero-shot generalizability, i.e., inference on an unseen system without any further training.
>
> **W3. There are several unclear points as follows:**
>
> **This study is formulated based on the Lagrangian form. What is the reason for choosing the Lagrangian form? Also, are the ideas in this study applicable to Hamiltonian form?**
>
> *Response:* In the present work, the study is formulated in the standard Newtonian form for dynamical systems. Indeed, we show that the formulation is equivalent to Lagrangian (and equivalently to Hamiltonian as well) mechanics. However, it should be noted that Lagrangian dynamics is traditionally applied in systems with conservative forces, i.e., where a Lagrangian is well-defined. However, the Eq (2) and the formulation is generic and applicable to model any dynamical system irrespective of whether the system has a well-defined Lagrangian or not. Examples of system where a well-defined Lagrangian may not exist include that of a colloidal system undergoing brownian motion.
>
> To address this comment, new text is added in Sec. 2 as follows.
>
> > We note that Lagrangian dynamics is traditionally applied in systems with conservative forces, i.e., where a Lagrangian is well-defined. However, Eq. 2 is generic and can be applied to model any dynamical system irrespective of whether the system has a well-defined Lagrangian or not. Examples of system where a well-defined Lagrangian may not exist include that of a colloidal system undergoing brownian motion.

---

> > ### Author Response · Authors · 2022-11-15
> > **Part 2**
> >
> > **Can you also briefly tell me about the advantages of studying physical systems based on GNN?**
> >
> > *Response:* The key advantage of using a GNN is *inductive modeling*. Due to using a GNN (GNode in our context), the number of parameters of the model is independent of the size of the system (the number of particles). This enables inductive modeling, i.e., zero-shot generalizability to arbitrary sized systems. The number of parameters are fixed, since the transformation operations conducted at each node (particle) of the graph are identical (Eqs. 9-13), which is to aggregate messages from each neighboring node and then passing it through an MLP. In a generic neural ODE, the parameter set grows with the size of the system and hence it cannot be applied on an $x$-sized system if trained on an $y$-sized system, where $x\neq y$. This happens since a different parameter set is learned for each particle.
> >
> > GNN also enables more robust training. Specifically, since the parameter set of NODE grows with system size, it is increasingly hard to train, both in terms of number of epochs and the volume of training data required, for large systems. GNode does not suffer from this issue since the parameter set is independent of system size.
> >
> > **The more inductive biases you consider, the more robust you will be against sparsity and noise in your data. If you have any views or experimental results, please let me know.**
> >
> > *Response:* This observation is correct. We have conducted new experiments to demonstrate this property. The discussion is available in Sec 4.6. In addition, we also show that the inductive biases lead to more more data-efficient training, i.e., we are able to learn from less training samples (App. F).
> >
> > **Appeal to the reviewer:** We thank the reviewer for the constructive criticism of our work. We have incorporated all of them in the revised version. We hope that the new experiments and additional explanations have convinced you of the merits of our work. If the reviewer agrees, we humbly request to raise  the rating accordingly.

---

> ### Author Response · Authors · 2022-11-17
> **Looking forward to feedback from Reviewer okft**
>
> Dear Reviewer,
>
> We thank you for providing critical and constructive feedback. To address the concerns raised by the reviewer, we have carried out several additional experiments and modifications which have now significantly improved the quality of the manuscript. Specifically, the major changes made in response to the comments by the reviewer are outlined below.
>
> 1. **New baseline** SymODE-Net is added for 3-,4-,5-spring and pendulum systems,
> 2. Additional experiments and a new section on **robustness against noise**, where the performance of different models for noisy data with varying noise percentages is critically evaluated,
> 3. **Additional references and text** have been added to improve the explanation and refer appropriate related work,
>
> With these additional experiments and improved explanations, we hope the reviewer now finds the manuscript significantly improved and acceptable. Please do let us know if there are any additional concerns.
>
> Looking forward to your response.
>
> Thank you,
>
> Authors

---

> ### Author Response · Authors · 2022-11-18
> **Eagerly awaiting feedback from Reviewer okft**
>
> Dear Reviewer,
>
> Since we are into the last day of author-reviewer discussions, we hope to engage into meaningful discussions regarding the comments and response. As noted below and in the general response, we have added several additional experiments and made significant revisions to the paper to address all the concerns raised. If there are any outstanding concerns, please do let us know. Otherwise, we would really appreciate if you raise your score.
>
> Looking forward to your response!
>
> Authors

---

> ### Author Response · Authors · 2022-11-19
> **Keenly awaiting post-rebuttal feedback from Reviewer okft**
>
> Dear Reviewer,
>
> Since we are into the last few hours of author-reviewer discussions, we keenly await your feedback on the additional experiments and modifications to address your comments. We believe we have addressed all the concerns raised by you rigorously. If there are any outstanding let us know.
>
> Based on our response, one reviewer has raised the score and another has provided a positive feedback. We look forward to your response and appreciate any feedback.
>
> Thank you,
>
> Authors

---

> > ### Comment · Reviewer_okft · 2022-11-19
> > **Thanks for your response**
> >
> > Sorry for the late reply. I have read your response. I think my concerns have been largely addressed. I will consider raising the score in later discussion phase.

---

> > > ### Author Response · Authors · 2022-11-19
> > > **Thank you!**
> > >
> > > Thank you for the response and positive feedback! We appreciate it!

---

### Official Review · Reviewer_gQW5 · 2022-10-25

**Confidence:** 3
**Correctness:** 3
**Technical Novelty And Significance:** 2
**Empirical Novelty And Significance:** 2
**Recommendation:** 6

**Clarity, Quality, Novelty And Reproducibility:**

- Clarity: The paper is well-written and easy to read.
- Quality: The method is straightforward and the setup of ablation study of inductive biases is justified.
- Novelty: The proposed method seems to be very similar to existing ones in the literature.
- Reproducibility: The details in the appendix seems to be enough for reproducibility.

**Strength And Weaknesses:**

## Strength
- The paper is clear and easy to follow.

## Weakness
- The motivation in the abstract is weak as generalizing to arbitrary-sized particle systems using graph neural networks with ODEs are already done by LGN/HGN as mentioned.
- The differences between the proposed method and LGN/HGN are not clear. It looks to me that the proposed method slightly generalize LGN/HGN by implementing different inductive biases. If this is the case, the novelty is limited.

## Questions
- Can you clarity the difference between MCGnode and LGN/HGN (in terms of inductive bias)?
- In the result section it point that MCGnode outperforms both LGN and HGN (figure 5) without any explanation on potential reasons/conjectures. Why this happens?
    - In general, the results section should not be a simple number comparison. You should draw some conclusion from the comparison.
- How are the different inductive biases studied here similar to [1]? Some comparisons could be useful as [1] is also a recent work with similar focus.

[1] Xu, K., Srivastava, A., Gutfreund, D., Sosa, F., Ullman, T., Tenenbaum, J. and Sutton, C., 2021. A Bayesian-symbolic approach to reasoning and learning in intuitive physics. Advances in Neural Information Processing Systems, 34, pp.2478-2490.

**Summary Of The Paper:**

The paper proposes to combine graph neural networks and neural ordinary differential equations (ODEs) to make the learned model generalize to particle systems with arbitrary sizes. It further studies how different inductive biases that could be embedded into the framework affect the performance. On two synthetic datasets, some benefits of the proposed model over two strong baselines are demonstrated.

**Summary Of The Review:**

It looks to me that the proposed method itself is not novel enough and the main contribution is the specific implementation to achieve several inductive biases and an ablation study on them. If this is not correct, the author could clarify this aspect by answering my questions in earlier section.

---

> ### Author Response · Authors · 2022-11-15
> **Response to Reviewer gQW5: part 1**
>
> **W1. The motivation in the abstract is weak as generalizing to arbitrary-sized particle systems using graph neural networks with ODEs are already done by LGN/HGN as mentioned. The differences between the proposed method and LGN/HGN are not clear. It looks to me that the proposed method slightly generalizes LGN/HGN by implementing different inductive biases. If this is the case, the novelty is limited.**
>
> *Response:* First, we humbly point out the original HGN is published in a non-archival workshop (see: [ML4PS, NeurIPS 2019](https://www.wikicfp.com/cfp/servlet/event.showcfp?eventid=93080)). This work focused only on spring systems and not on pendulum-like systems that behave under the influence of external fields. Further, there is no LGN published in any archival venues that replicates physical systems, to the best of our knowledge. However, LGN is a minor modification over HGN where instead of Hamiltonian, the Lagrangian is computed by the neural network. Despite proper publication/documentation of HGN and LGN in archival venues, we used these models as baselines in our work as we felt there is limited novelty in extending LNNs and HNNs to simply a graph-based architecture. To address this gap, we have evaluated LGN and HGN carefully, while comparing it with MCGNode in terms spring and pendulum systems, zero-shot generalizability, and energy, rollout and momentum error.
>
> Second, the present work is not a slight generalization over LGN or HGN. In the present work, we carefully investigate how appropriate inductive biases can be added to a Node-based system to achieve the same or better accuracy than LGN or HGN. This is important as LGN and HGN are known to be computationally expensive due to the gradient operations involved in their formulation. Specifically, while in GNode(and MCGNode), the predicted acceleration is the direct output of the graph neural network, in LGN (or HGN) the output is the Lagrangian (or Hamiltonian) and further differentiation of the Lagrangian (or Hamiltonian) provides acceleration. These additional computational steps make LGN or HGN computationally expensive. GNode directly predicts the acceleration since it learns to approximate the system-dynamics function $F(q_t,\dot q_t)$. Furthermore, when the appropriate inductive biases are incorporated, such as in CDGNode, the resulting Graph Neural Network is dramatically different from LGN and HGN in terms of the objective function, architecture, number of model parameters and the resulting computations.
>
> To clearly bring out these differences, we have introduced the following changes:
>
> * **Differentiation with LGN and HGN:** We explcitly discuss the differences of GNode with LGN and HGN in Sec 4.1.
> * **Impact of inductive biases on performance and data efficiency:** We have expanded our discussion on the inductive biases and the resulting architecture. In addition to providing superior accuracy, we now also demonstrate that the inductive biases enable learning from significantly lower volume training data App. F and Fig. 12.
> * **Computational Efficiency:** We empirically establish the computational superiority of the MCGNODE over HGN and LGN in terms of training and inference times (Tables 1 and 2 in the appendix and App F.). In spring systems, for example, MCGNODE is $\approx 9$ times faster than HGN and $\approx 90$ times faster than LGN.
> * **Robustness to noise:** We empirically establish that MCGNode is significantly more robust to noise when compared to GNODE, HGN and LGN (See Sec 4.6 and Fig. 7). This further substantiates the importance of designing appropriate inductive biases, which is one of the core contributions of our work.
>
> On the aspect of novelty, zero-shot generalizability is just one contribution but not the central one:
>
> * **How do you incorporate topology-aware semantics on neural ODEs to make them inductive?** As already discussed above, this does not involve a simple generalization of LGN or HGN for ODEs.
>
> * **How do you inject inductive biases to model constraints?** We rigorously establish that the proposed inductive biases result in conservation of momentum (Theorem 1). We also introduce judicious changes to the neural architecture that enable decoupled modeling of internal and external forces (See CDGNode) and thereby ensure better coherence between semantics of the underlying dynamics with the learning process.

---

> > ### Author Response · Authors · 2022-11-15
> > **part 2**
> >
> > (Continuation of W1..)
> >
> > * **Empirical demonstration of zero-shot generalizability and impact of inductive biases**: In the literature, zero-shot generalizability has not been demonstrated either by LGN or HGN (although they are theoretically capable of doing so). This work provides a more rigorous treatment to this important aspect and thereby leads to a more comprehensive understanding of the limitations and strengths of the various paradigms for learning dynamics of physical systems. In addition, we also systematically study the impact of the various inductive biases in learning dynamics, which is potentially of value towards more systematic design of neural architectures for modeling dynamics of physical systems.
> >
> > * **Empirical demonstration of data-efficiency**: We demonstrate that MCGNode is able to learn from significantly smaller amounts of data than LGN and HGN by almost 2-3 orders of magnitude and with much higher performance.
> >
> > **W2. Can you clarify the difference between MCGnode and LGN/HGN (in terms of inductive bias)?**
> >
> > *Response:* We have addressed the differences of GNode with LGN and HGN already in W1. MCGnode incorporates three inductive biases to model various physical constraints. All three biases are novel, non-trivial, and are not part of LGN or HGN. The specifics are as follows:
> >
> > * **Newton's third law (Sec 3.4)**: We enforce the bias that the force exerted on a node (particle) is symmetric to the force the node exerts on its neighbors. We rigorously prove that this leads to conservation of momentum. This is an original contribution of the paper. LGN or HGN *does not* enforce this bias and hence does not guarantee momentum conservation. This is demonstrated in Fig. 4, where MCGNode clearly outperforms LGN and HGN.
> >
> > * **Decoupling of internal and external forces (Sec 3.3)**: In a physical system, internal forces arise due to interactions between constituent particles. On the contrary, the body forces are due to the interaction of each particle with an external field such as gravity or electromagnetic field. To make our architecture consistent with this semantic, we learn two representations for each node; one through a GNN that models internal topology-dependent forces, and a global embedding through an MLP modeling forces emanating from external fields. We have now made this more explicit in Fig. 1 depicting our neural architecture. This decoupling is not present in either HGN or LGN. Due to the absence of this bias, HGN and LGN perform poorly for pendulum-like systems where external field plays a dominant role as shown in Fig. 3.
> >
> > * **Decoupling dynamics and constraints (Sec 3.2):** Instead of directly predicting the acceleration of each node $v$, we predict the force on $v$ and then the acceleration is computed from this force using Eq. 5. This enables us to isolate the effect of imposing constraints explicitly on the learning, and the performance of GNode. Eq. 5 is specific to neural ODE based modeling and does not apply as it is to LGN or HGN.
> >
> > As demonstrated by the experiments, the inclusion of these additional inductive biases:
> > 1. enhances the performance of MCGNode significantly leading to lower error than LGN and HGN in most cases (Sec. 4.3, Figs. 4, 5),
> > 2. makes the forward trajectory more meaningful physically by making it respect the constraints, momentum conservation, and Newton's third law (Sec. 4.3, Fig. 4),
> > 3. makes MCGNode more robust against noise (Sec. 4.6, Fig. 7,)
> > 4. makes MCGNode significantly more efficient in terms of the number of data-points required for training (App. F, Fig. 12)

---

> > > ### Author Response · Authors · 2022-11-15
> > > **part 3**
> > >
> > > **W3. In the result section it point that MCGnode outperforms both LGN and HGN (figure 5) without any explanation on potential reasons/conjectures. Why this happens?**
> > >
> > > *Response:* The primary reasons why MCGNode outperforms LGN and HGN are due to the additional inductive biases encoded in the MCGNode as outlined in the previous comment. Specifically, the reasons are as follows.
> > > 1. **Expicit constraints:** Explicit constraints as applied in MCGNode significantly reduce the error, while the absence of explicit constraints in LGN or HGN results in poorer performance for systems with constraints such as pendulums.
> > > 2. **Decoupling dynamics constraints:** Decoupling the dynamics and constraints enhances the performance of MCGNode in systems such as pendulums or systems with external drag forces, where external fields are present.
> > > 3. **Momentum conservation:** Enforcing momentum conservation explicitly leads to improved performance of MCGNode in systems where external forces are absent such as an $n$-spring system.
> > >
> > > Altogether, the carefully constructed inductive biases addresses the gap in the state-of-the-art models such as LGN, HGN, and GNodes, thereby ensuring significantly better models that are physically reasonable in addition to being more accurate and efficient.
> > >
> > > To address this comment, new text is added in Sec. 4.3 as follows:
> > > > The superior performance of MCGNode could be attributed to the carefully constructed inductive biases, namely, (i) explicit constraints---improves data efficiency and reduces the error in pendulum systems, (ii) decoupling internal and body forces---leads to superior performance in pendulum systems which act under gravity, (iii) momentum conservation---leads to superior performance in spring systems by enforcing an additional conservation law strongly. We note that these inductive biases may also be adapted suitably for LGN and HGN and thereby improve their performance. Thus, the contribution of our work is not the neural architecture in isolation, but the core design principles that are generic enough to be abstracted beyond neural ODEs.
> > >
> > > **W4. In general, the results section should not be a simple number comparison. You should draw some conclusion from the comparison.**
> > >
> > > *Response:* As mentioned in detail above, additional text and analysis has been added to draw the conclusions on which the inductive biases enable superior performance of MCGNode in comparison to LGN and HGN.
> > >
> > > **W5. How are the different inductive biases studied here similar to [1]? Some comparisons could be useful as [1] is also a recent work with similar focus.**
> > >
> > > **[1] Xu, K., Srivastava, A., Gutfreund, D., Sosa, F., Ullman, T., Tenenbaum, J. and Sutton, C., 2021. A Bayesian-symbolic approach to reasoning and learning in intuitive physics. Advances in Neural Information Processing Systems, 34, pp.2478-2490.**
> > >
> > > *Response:* We thank the reviewer for pointing to the relevant reference. The suggested work is similar in spirit that it infuses priors to enhance predictions of physical systems. In the work, while the access to Newtonian mechanics is assumed, hand-crafted symbolic inputs are used in the model as part of pre-processing steps. In the present work, the dynamics are learned directly from the trajectory based on the observable quantities, namely, velocity and positions. Further, the inductive biases in the present case are carefully constructed to make the learning more efficient, and the resulting trajectory more realistic from a physical perspective.
> > >
> > > We have now cited and discussed this work in Sec. 1.
> > >
> > > **Appeal to the reviewer:** We thank the reviewer for the constructive comments on our work. We have incorporated all of them in the revised version. We hope the reviewer now finds our paper improved. If the reviewer agrees, we humbly request to alter the rating accordingly.

---

> > > > ### Comment · Reviewer_gQW5 · 2022-11-19
> > > > **Thanks for the response**
> > > >
> > > > Thanks for explaining the novelty as well as why the proposed method tends to work better than competing approaches.
> > > > The revised draft looks better now and I raised my score accordingly.

---

> > > > > ### Author Response · Authors · 2022-11-19
> > > > > **Thank you!**
> > > > >
> > > > > We thank the reviewer for the constructive feedback and for the positive response. Thank you!

---

> > ### Comment · Reviewer_gQW5 · 2022-11-19
> > **Re. GNode vs LGN/HGN**
> >
> > Is it fair to say that GNode deals with (or is restricted to) Newtonian dynamics?

---

> > > ### Author Response · Authors · 2022-11-19
> > > **Response regarding GNode**
> > >
> > > Thank you for engaging in the discussion!
> > >
> > > Since the premise (governing differential equation) of the GNode is based on the Newton’s second law, it is reasonable to say that GNode follows Newtonian dynamics. We shall include a sentence in Sec. 2 of the updated manuscript to clarify this point!

---

> ### Author Response · Authors · 2022-11-17
> **Looking forward to feedback from Reviewer gQW5**
>
> Dear Reviewer,
>
> We thank you for taking the time to provide critical comments which has significantly improved the manuscript quality. Since we are in the last two days for author-reviewer discussion period, we hope to engage in a discussion and improve the paper to the best extent possible.
>
> As outlined below, we have now clearly highlighted the novelty of the manuscript and provided in-depth discussions on the intuitions and nature of the inductive biases employed. In addition, we have also included detailed discussions on why these inductive biases result in superior performance. To this extent, we have also performed several additional experiment such as **robustness against noise**, additional baseline **SymODE-Net**, **data efficiency**, and **computational efficiency**.
>
> With these additional experiments and improved explanations, we hope the reviewer now finds the manuscript acceptable. If there are any outstanding concerns, we request the reviewer to raise those.
>
> Looking forward to your response.
>
> Thank you,
>
> Authors

---

> ### Author Response · Authors · 2022-11-18
> **Eagerly waiting for your feedback on our revisions.**
>
> Dear Reviewer,
>
> Since we are entering the last few hours of incorporating any further revisions in our work, we are eagerly awaiting your feedback and whether you feel the need to address any further outstanding concerns. As already highlighted, we have conducted additional experiments to substantiate the usefulness of the inductive biases, as well as incorporate more depth in the differentiation with respect to LGN and HGN.
>
> We hope with the new revisions, the reviewer finds our work acceptable.
>
> Looking forward to your feedback,
>
> regards,
>
> Authors

---

### Official Review · Reviewer_UJsG · 2022-11-04

**Confidence:** 4
**Clarity, Quality, Novelty And Reproducibility:** Please see above.
**Correctness:** 3
**Technical Novelty And Significance:** 3
**Empirical Novelty And Significance:** 3
**Recommendation:** 8

**Strength And Weaknesses:**

Strengths:

1. Indeed, GNNs have been explored widely for physical modeling. This paper applies GNN to characterize the terms of the second-order ODE in Eq. 5, which has contained sufficient terms that reflect conservative/non-conservative forces, internal/external forces, and constraints. This is novel.

2. The authors develop a series of models that take into account different physics priors. The discussion on the relation with Newton’s third law is interesting.

3. Experiments are conducted compactly but convincingly. It is able to justify the validity of the proposed models, particularly CGNODE, CDGNODE, MCGNODE.


Questions:

Below, I raise some questions (not necessarily weakness) and hope the authors can provide answers to them.

1. Is there any gap like assumption or relaxation between Eq. (2) and Lagrangian dynamics in Eq. (1). If so, please specify.

2. How to compute the Coriolis-like forces in your models, and how to make M inversible in Eq. (4).

3. How does the derivation of lambda by (4) differ from Eq. (7) in the paper [Simplifying Hamiltonian and Lagrangian Neural Networks via Explicit Constraints].

4. N_i = Gamma_i =0 in all your experiments?

5. For your models, CGNODE, CDGNODE, MCGNODE, the inductive biases are added independently, or  enforced upon each other? Why not propose a eventual model that considers all biases?

6. For the baselines LGN and HGN, does it share the same GNN backbone as GNODE?

7. What is the implementation of NODE?

8. From Figure 2, it is hard to verify that GNODE is clearly better than NODE. If so, please modify the presentation in the paragraph of “Impact of topology”.

9. Eq. (3), there is a typo. There is no constraint error to check if the proposed model can really permit the constraint of the systems.

Suggestions:

1. Section 3.1 is not the central contribution. The authors are suggested to introduce more on Section 3.2-3.4 including moving their details and illustrated figures from appendix to the main body.

2. The neural network in Figure 1 is just an MLP and unable to show the case of GNN or message passing.

3. It will be interesting to visualize the constraint errors, the difference between the internal force and external force distribution of the learned model.

4. There are recent papers that focus on enhancing symmetry on physical modeling, leading to the application of equivariant GNNs; examples include EGNN, GMN that also involves constraints, SGNN that considers external force field. I am not asking the authors to conduct experimental comparisons wit them, but rather encourage the authors to add some related discussion with the methods in that domain, since symmetry is actually a central topic in physics. For example, if we enforce the Lagrangian to be invariant w.r.t. space translation/rotation, we will also derive the Newton’s third law just based on Lagrangian dynamics. This point is easily justified, by for example checking the Eq. (5) in GMN, where the force message f_ij on each edge is of the same orientation of the relative position x_ij of each two nodes, indicating f_ij=-f_ji.


[EGNN] E(n) Equivariant Graph Neural Networks, ICML 2021.
[GMN] Equivariant graph mechanics networks with constraints, ICLR 2022.
[SGNN] Learning Physical Dynamics with Subequivariant Graph Neural Networks, NIPS 2022.


**Summary Of The Paper:**

This paper proposes to model the second-order ODE of multi-body dynamical systems by using Graph Neural Networks (GNNs). Different from previous LGN or HGN that directly predicts the Lagrangian/Hamiltonian of the system, this paper explicitly involves the inductive biases of the constraints, internal-external forces, and conservation of Newton’s third law.  Experiments are performed to support the benefit of considering the above biases, and the generalizability to unseen systems of large sizes.

**Summary Of The Review:**

I generally enjoy reading this paper and accept the ideas. I have some concerns on the writing and the experiments.

---

> ### Author Response · Authors · 2022-11-15
> **Response to Reviewer UJsG: Part 1**
>
> We thank the reviewer for the positive comments. Please find the point-by-point response below to each of the comments.
>
> **Is there any gap like assumption or relaxation between Eq. (2) and Lagrangian dynamics in Eq. (1). If so, please specify.**
>
> *Response:* Lagrangian dynamics is traditionally applied to systems with conservative forces, that is, where a Lagrangian is well-defined. However, Eq. (2) is generic and can be applied to model any dynamical system irrespective of whether the system has a well-defined Lagrangian or not. Examples of systems where a well-defined Lagrangian may not exist include that of a colloidal system undergoing Brownian motion. Even in such cases, the dynamics as defined by Eq. (2) stands valid.
>
> To clarify this point, additional text has been added to Sec. 2 following Eq. (2).
>
> **How to compute the Coriolis-like forces in your models, and how to make M inversible in Eq. (4).**
>
> *Response:* Since the present work focuses on particle-based systems, Coriolis-like forces always go to zero due to the absence of terms having both $q$ and $\dot{q}$. This can be explained based on the energetic contribution to each particle, where kinetic energy will be a function of $\dot{q}$ only and potential energy a function of $q$ only. This term will be non-zero in the case of rigid body dynamics, where the kinetic energy can be a function of both $q$ and $\dot{q}$. Hence, computation of the Coriolis-like forces is not relevant for the present work. In any case, this force can be performed by employing the additional relationship between $C$ and $N$ based on Newtonian dynamics, that is, $C(q,\dot{q})=\partial N/ \partial \dot{q}$.
>
> Similarly, to compute the mass matrix $M$, we invoke the fact that in particle-based systems the mass matrix is diagonal with each of the diagonal representing the mass associated with a degree of freedom. This ensures that the mass matrix is invertible. A similar approach has been used earlier to simplify the learning, for instance, see Ref. Finzi et al.
>
> Finzi, M., Wang, K.A. and Wilson, A.G., 2020. Simplifying Hamiltonian and Lagrangian neural networks via explicit constraints. Advances in neural information processing systems, 33, pp.13880-13889.
>
> **How does the derivation of lambda by (4) differ from Eq. (7) in the paper [Simplifying Hamiltonian and Lagrangian Neural Networks via Explicit Constraints].**
>
> *Response:* The derivation of $\lambda$ is similar to the approach employed in the above-mentioned paper (Finzi et al). We clarify the approach in the present paper in the context of Newtonian dynamics. This is included in the text for the sake of completeness.
>
> **N_i = Gamma_i = 0 in all your experiments?**
>
> *Response:* We thank the reviewer for raising this point. To demonstrate the capability of MCGNode to learn the dynamics of systems with drag and external forces, we performed additional experiments on systems with linear drag and velocity. Specifically, we evaluated the performance of MCGNode on a 5-spring system with linear drag (that is, drag force given by $-0.1\dot{q}$ and a non-zero external force). We note that the MCGNode clearly outperforms Node.
>
> To address this point, new results are added to the Appendix (App. E) with references from Sec 4.2 and Sec 3.2.
>
> **For your models, CGNODE, CDGNODE, MCGNODE, the inductive biases are added independently, or enforced upon each other? Why not propose a eventual model that considers all biases?**
>
> *Response:* They are enforced upon each other. Thus, MCGNode is the eventual model that considers all the biases. This also outlines why MCGNode exhibits superior performance in comparison to other models. To make this explicit, we have added the following paragraph in Sec 3.1.
>
> > GNode learns the dynamics directly without decoupling the constraints and other components governing the dynamics from the equation $\ddot q = \hat{F}(q,\dot q, t)$. Next, we analyze the effect of providing the constraints explicitly in the form of inductive biases and enhance the performance of GNode. In the subsequent sections, we iteratively stack inductive biases on top of GNode. Thus, the final proposed architecture, MCGNode, is the eventual model that considers all biases.

---

> > ### Author Response · Authors · 2022-11-15
> > **Part 2**
> >
> > **For the baselines LGN and HGN, does it share the same GNN backbone as GNODE?**
> >
> > *Response:* The baselines LGN and HGN were trained using a full graph architecture as employed in the original reference (see Refs. Battaglia et al, and Sanchez-Gonzalez et al., full citations below). However, in order to address the comment, we have performed additional experiments of LGN and HGN using our GNN. This version of LGN and HGN perform significantly worse. The results are available at App. C with a reference from Sec 4.1.
> >
> > 1. Battaglia, P.W., Hamrick, J.B., Bapst, V., Sanchez-Gonzalez, A., Zambaldi, V., Malinowski, M., Tacchetti, A., Raposo, D., Santoro, A., Faulkner, R. and Gulcehre, C., 2018. Relational inductive biases, deep learning, and graph networks. arXiv preprint arXiv:1806.01261.
> > 2. Sanchez-Gonzalez, A., Bapst, V., Cranmer, K. and Battaglia, P., 2019. Hamiltonian graph networks with ODE integrators. arXiv preprint arXiv:1909.12790.
> >
> > **What is the implementation of NODE?**
> >
> > *Response:* We use the implementation provided in Gruver et al., 2022. The full citation is provided below. We mention this in Baselines para of Sec 4.1.
> >
> > *Nate Gruver, Marc Anton Finzi, Samuel Don Stanton, Andrew Gordon Wilson, Deconstructing the Inductive Biases of Hamiltonian Neural Networks, ICLR 2022.*
> >
> >
> > **From Figure 2, it is hard to verify that GNODE is clearly better than NODE. If so, please modify the presentation in the paragraph of “Impact of topology”.**
> >
> > *Response:* The performance of NODE and GNODE are indeed similar. Note that we are not claiming GNODE to be better than NODE in terms of accuracy. Rather, we showcase the inductive ability of GNODE. Specifically, while NODE is trained and tested on each system separately (3,4,5 sized spring and pendulum systems), GNODE is trained only on 3-pendulum and 5-spring systems and performs zero-shot inference on the unseen systems. Yet, the performance of GNODE is as good as NODE, which highlights its ability to generalize to arbitrary sized systems. We have now emphasized this point further in Sec 4.2. Note that MCGNode is strictly better than NODE due to the injection of additional biases modeling constraints.
> >
> > Additional text is added to Sec 1, Sec. 4.2 and App D to address this comment.
> >
> > **Eq. (3), there is a typo. There is no constraint error to check if the proposed model can really permit the constraint of the systems.**
> >
> > *Response:* Eq. (3) is obtained directly by rearranging Eq. (2) based on the physical system. Further, in the present work, we are not learning the constraints, but directly employing it as a bias based on the knowledge of the systems. As such, there is no constraint error. Indeed, learning the constraints directly can be a future extension of the work.
> >
> > We now acknowledge this explicitly as a future direction in Sec 5.
> >
> > **Suggestions:**
> >
> > **Section 3.1 is not the central contribution. The authors are suggested to introduce more on Section 3.2-3.4 including moving their details and illustrated figures from appendix to the main body.**
> >
> > *Response:* As suggested, we have shortened Section 3.1, and included more details of the inductive biases described in Sec 3.2-3.4, including the architecture of the best model MCGNode. Further, we have now moved the details of the graph architecture to Appendix (see App. A).
> >
> > **The neural network in Figure 1 is just an MLP and unable to show the case of GNN or message passing.**
> >
> > *Response:* We have now updated this figure to explicitly separate out the MLP and GNN components. Further, we have added a new figure in Appendix to explain the GNode architecture (see App. A).
> >
> > **It will be interesting to visualize the constraint errors, the difference between the internal force and external force distribution of the learned model.**
> >
> > *Response:* Indeed, there would be differences (errors) in the internal force learned by the model and the actual internal forces. This is captured in a cumulative fashion to some extent in the momentum error since force is the rate of change of momentum. To address this comment, we plot the predicted internal force vs actual internal force for each of the particles in the spring and pendulum systems by GNode, MCGNode, HGN and LGN in Appendix G.
> >
> > Note that there are no constraint errors in the MCGNode, since the constraints are explicitly encoded through the $A^T \lambda$ term. However, the constraint forces can still exhibit error. To demonstrate this further, we evaluate the internal forces and constraint forces in spring and pendulum systems (see App. G) predicted by GNode, MCGNode, LGN, and HGN. We observe that MCGNode exhibits excellent agreement with the actual forces and is superior to GNODE. This substantiates the efficacy of the inductive biases.

---

> > > ### Author Response · Authors · 2022-11-15
> > > **Part 3**
> > >
> > > **There are recent papers that focus on enhancing symmetry on physical modeling, leading to the application of equivariant GNNs; examples include EGNN, GMN that also involves constraints, SGNN that considers external force field. I am not asking the authors to conduct experimental comparisons with them, but rather encourage the authors to add some related discussion with the methods in that domain, since symmetry is actually a central topic in physics. For example, if we enforce the Lagrangian to be invariant w.r.t. space translation/rotation, we will also derive the Newton’s third law just based on Lagrangian dynamics. This point is easily justified, by for example checking the Eq. (5) in GMN, where the force message $f_{ij}$ on each edge is of the same orientation of the relative position $x_{ij}$ of each two nodes, indicating $f_{ij}=-f_{ji}$.**
> > >
> > > 1. **[EGNN] E(n) Equivariant Graph Neural Networks, ICML 2021.**
> > >
> > > 2. **[GMN] Equivariant graph mechanics networks with constraints, ICLR 2022.**
> > >
> > > 3. **[SGNN] Learning Physical Dynamics with Subequivariant Graph Neural Networks, NeurIPS 2022.**
> > >
> > > *Response:* We thank the reviewer for suggesting some of these interesting recent works. We have now cited and discussed this in detail in Sec 1.
> > >
> > > The observation on deriving Newton's third law by enforcing invariance on the Lagrangian w.r.t. Space translation/rotation is correct. We have acknowledged this explicitly. In general, while the focus of our work is on designing useful inductive biases, the referred works are focused towards better representation learning.
> > >
> > >
> > > **Appeal to the reviewer:** We thank the reviewer for the constructive comments on our work. We have incorporated all of them in the revised version. We hope the reviewer now finds our paper improved. If the reviewer agrees, we humbly request to raise the rating accordingly.

---

> ### Author Response · Authors · 2022-11-17
> **Looking forward to feedback from Reviewer UJsG**
>
> Dear Reviewer UJsG,
>
> We thank you for taking the time to provide constructive comments which have significantly improved the quality of the manuscript. Since we are in the last two days for author-reviewer discussion period, we hope to engage in a discussion and improve the paper to the best extent possible. Specifically, the major changes made in response to the comments by the Reviewer are outlined below.
> 1. Additional experiments with **drag and external force** to demonstrate the capability of MCGNode to learn the dynamics in these systems,
> 2. Additional experiments on **LGN and HGN** with GNN used in the GNode as the backbone.
> 3. Visualisation and comparison of **internal and constraint forces** in the spring and pendulum systems.
> 4. **Updated Figure 1** to explain the GNode and GNN architecture.
>
> With these additional experiments and improved explanations, we hope we have addressed all the concerns raised by the reviewer.
> If there are any outstanding concerns, we request the reviewer to please raise those. Otherwise, we would really appreciate if the reviewer can raise the score.
>
> Looking forward to your response.
>
> Thank you,
>
> Authors

---

> > ### Comment · Reviewer_UJsG · 2022-11-18
> > **More explanations are needed.**
> >
> > Thank the authors for the feedbacks. They help address most concerns. Here is one additional comment.
> >
> > Have you tried more complex systems, such as the cases used in GNS paper? There is no other inductive bias used in GNS besides the graph modeling. However, the experiments on WATER-3D, GOOP3D and SAND-3D datasets are fancy to show the power of GNN for physical simulation.  I will raise my score if the authors can comment and particularly experiment on this point.
> >
> > Minor issue: The years of the newly added references are mistaken. Please correct them.

---

> > > ### Author Response · Authors · 2022-11-20
> > > **Results on 3D solid system**
> > >
> > > We thank the reviewer for the additional comments and engaging with us in a discussion for further improving our work.
> > >
> > > Indeed, we have evaluated the performance on complex systems as discussed in Sec 4.5, where we demonstrate the superior performance of MCGNode over GNode for the deformation of a 3D solid system subjected to isotropic compression. To compare the performance with GNS, we have now added it as an additional baseline. The tables below demonstrate that GNS performs inferior to MCGNode. Further, the complex nature of deformation can be visually studied in the videos provided at our anonymous [code-repo link](https://anonymous.4open.science/r/graph_neural_ODE-8B3D) (See Sec 4, first paragraph). For better comparison, the trajectories from ground truth and MCGNODE are simulated together with the square (2D)/cube(3D) icon representing the ground truth trajectory and circle (2D)/sphere(3D) representing the predicted trajectory.
> > >
> > > Although, we tried to include other 3D systems as the reviewer suggested, due to paucity of time we were unable to include it. However, we believe the present experiment indeed demonstrates the power of MCGNode to model complex systems. The results on other systems shall be shared as a comment in the OpenReview as soon as we get those.
> > >
> > > Regarding the minor comment, we will correct the incorrect dates. Thanks for pointing them out.
> > >
> > > Thank you!
> > >
> > > -----------------------------------------
> > >
> > > # Momentum Error
> > >
> > > | Time(s)	| GNS	| GNODE	| MCGNODE |
> > > | ----  | ---   | ----- | ------- |
> > > | 0.00	| 0.74	| 0.92	| 0.26 |
> > > | 1.00	| 0.88	| 1.00	| 0.28 |
> > > | 2.00	| 0.90	| 1.00	| 0.30 |
> > > | 3.00	| 0.89	| 1.00	| 0.36 |
> > > | 4.00	| 0.81	| 1.00	| 0.52 |
> > > | 5.00	| 0.84	| 1.00	| 0.80 |
> > > | 6.00	| 0.88	| 1.00	| 0.81 |
> > > | 8.00	| 0.88	| 1.00	| 0.76 |
> > > | 9.00	| 0.85	| 1.00	| 0.75 |
> > > | 9.90	| 0.84	| 1.00	| 0.70 |
> > >
> > > # Rollout Error
> > >
> > > | Time(s)	| GNS	| GNODE	| MCGNODE |
> > > | ----  | ---   | ----- | ------- |
> > > | 0.00	| 0.00	| 0.00	| 0.00 |
> > > | 1.00	| 1.00	| 0.96	| 0.01 |
> > > | 2.00	| 1.00	| 1.00	| 0.02 |
> > > | 3.00	| 1.00	| 1.00	| 0.04 |
> > > | 4.00	| 1.00	| 1.00	| 0.06 |
> > > | 5.00	| 1.00	| 1.00	| 0.06 |
> > > | 6.00	| 1.00	| 1.00	| 0.09 |
> > > | 7.00	| 1.00	| 1.00	| 0.12 |
> > > | 8.00	| 1.00	| 1.00	| 0.16 |
> > > | 9.90	| 1.00	| 1.00	| 0.21 |

---

> > > > ### Comment · Reviewer_UJsG · 2022-11-20
> > > > **The feedbacks are helpful!**
> > > >
> > > > Thank you! It looks great for the experimental comparison with GNS on the 3D solid systems. I suggest the authors to add these results into the revised version. By the way, the published year of GMN is 2022 not 2021.
> > > >
> > > > I have raised the score from 6 to 8.

---

> > > > > ### Author Response · Authors · 2022-11-20
> > > > > **Thank you!**
> > > > >
> > > > > We thank the reviewer for raising the score and for the constructive feedback. The results will be added to the final version of the manuscript.
> > > > >
> > > > > Thank you for pointing out the error in the GMN reference. We were unable to update the submission. This will be updated in the final version.
> > > > >
> > > > > Thank you!

---

### Author Response · Authors · 2022-11-15
**General comments to all the Reviewers**

We thank the reviewers for their insightful comments and suggestions. Please find a point-by-point response to the comments raised by the reviewers below. We have also updated the main manuscript and the appendix to address these comments. The changes made in the main manuscript are highlighted in **blue** color. The **major additional experiments** carried out and included in the updated manuscript are listed below.

1. **Noise experiments**: To analyse the effect of noise, we have now included additional experiments on noisy data (see Sec. 4.6 and Fig. 7)
2. **Systems with drag and external force**: To demonstrate the ability of MCGNode to model dissipative and non-conservative systems, additional experiments on systems with drag and external force is performed (see App. E, Fig. 11).
3. **Data efficiency**: The efficiency of MCGNode to learn the dynamics is compared with GNode, LGN, and HGN (see App. F, Fig. 12). We observe that MCGNode exhibits significantly efficient learning with small amounts of data.
4. **Training and simulation time**: The training and forward simulation time of MCGNode is compared with GNode, LGN and HGN (see App. F, Table 1, 2).
5. **Additional experiments on LGN and HGN baselines**: Additional experiments on LGN and HGN using the same graph architecture as employed in MCGNODE is performed for comparison (App. C, Fig. 9).
6. **Additional baseline**: We include an additional baseline SymODE-Net (SymODEN) and evaluated the performance in comparison to MCGNode (see App. J, Fig. 22). We observe that SymODEN is poorer than MCGNode suggesting the superiority of inductive biases employed in MCGNode.

---

### Author Response · Authors · 2022-11-17
**Looking forward to post-rebuttal feedback**

Dear Reviewers,

Thank you once again for all of your constructive comments, which have helped us significantly improve the paper! As detailed below, we have performed several additional experiments and analyses to address the comments and concerns raised by the reviewers.

Since the discussion phase is going to end in a day, we are eagerly looking forward to your post-rebuttal responses.

Please do let us know if there are any additional clarifications or experiments that we can offer. We would love to discuss more if any concern still remains. We appreciate your suggestions.

Thank you!

---

### Comment · Area_Chair_RUf5 · 2022-11-21
**Clarification of differences between versions**

Dear authors,

Thank you for your answers! The proposed algorithm has multiple versions and I personally would benefit from further clarification of the differences between them. I would appreciate if you could describe each version in the following way:

1. What are the inputs of each node/edge of a GNN?
2. How are the outputs of the GNN used to compute the loss function? What is the loss function?
3. If Eq. 5 is used to compute the loss function, which terms are assumed to be known and which terms are learned? What are the architectures used for the learned parts?
4. If some of the terms are assumed known, how are they set in the physical systems considered in the experiments?
5. What is the inductive bias introduced in this version?

Some questions about Figs. 1 and 8:
- Is Figure 8 correct? I think GNODE is supposed to output predicted accelerations.
- In Fig. 1, it is unclear which parts belong to CDGNODE and which to MCGNODE.
- Why do the topologies of the "graph representations of physical system" and the GNN differ?

Thank you!

---

> ### Author Response · Authors · 2022-11-22
> **Response to Area Chair RUf5**
>
> We thank the area chair for the careful reading and thoughout comments toward further improvement of the manuscript. Please find below the point-by-point responses to the comments raised.
>
> **The proposed algorithm has multiple versions and I personally would benefit from further clarification of the differences between them. I would appreciate if you could describe each version in the following way:**
> 1. **What are the inputs of each node/edge of a GNN?**
> 2. **How are the outputs of the GNN used to compute the loss function? What is the loss function?**
> 3. **If Eq. 5 is used to compute the loss function, which terms are assumed to be known and which terms are learned? What are the architectures used for the learned parts?**
> 4. **If some of the terms are assumed known, how are they set in the physical systems considered in the experiments?**
> 5. **What is the inductive bias introduced in this version?**
>
> *Response:* The differences between each of the architectures are as follows.
>  1. GNode:
>
> **Summary:** Graph version of Node that allows inductivity to unseen system sizes.
> * **Node inputs:** position ($q$), velocity ($\dot{q}$), and type ($t$)
> * **Edge inputs:** $w_{ij}=(q_i-q_j)$
> * **Node output:** acceleration of each particle ($\ddot{q}_i$). Note that we do not use Eq.(5) here to predict $\ddot{q}$. Instead, we directly predict the acceleration using $\ddot{q}_i=\hat{F}(q,\dot{q},t)$, where $\hat{F}$ is the approximate dynamics learned by the GNode. Here, predicting force and acceleration are equivalent problems since the acceleration is a simple scaling of the force.
> * **Edge output:** Nil.
> * **Architecture:** Fig. 8. We will clarify that the GNode directly predicts the acceleration and not the force although they are equivalent problems as detailed below.
> * **Loss function:** $\mathcal{L}= \frac{1}{n}\left(\sum_{i=1}^n \sum_{t=2}^\mathcal{T} \left(\ddot{q}_i^{\mathbb{T},t}-\hat{\ddot{q}}_i^{\mathbb{T},t}\right)^2\right)$
> * **Assumed knowns:** None, since we do not use Eq.(5) here.
> * **Inductive bias:** Modeling the physical system as a graph with the particles as nodes and connections between them as edges.
>  2. CGNode:
>
> **Summary:** Same as GNode except for the property that the neural network predicts the force from which acceleration is computed using Eq. (5).
> * **Node inputs:** Same as in GNode
> * **Node output:** The force $N_i$ on each particle $i$
> * **Edge outputs:** Nil, same as in GNode
> * **Architecture:** Fig. 8
> * **Loss function:** Same as in GNode. The acceleration is obtained using Eq.(5), where $N$ is obtained from the CGNode, $A(q)$ is assumed to be known, $M$ is the learnable diagonal mass matrix, $\Pi$ is the known external force, $C$ is 0, $\Upsilon$ is the learnable external drag. Note that unless specified, $\Pi$ and $\Upsilon$ is considered to be 0, that is, the system is not subjected to external force or drag. Details below.
> * **Assumed knowns:** (i) Governing equation for acceleration as given by Eq. (5). (ii) $k$ constraints (holonomic or Pfaffian) in the physical system represented by $A(q) \in \mathbb{R}^{k×D}$ based on the topology and nature (rigid/deformable)) of the system. For instance, the bar in the pendulum is rigid whereas the springs are deformable.
> * **Inductive bias:** Same as GNode + explicit constraints as given by $A(q)$ and acceleration as given by Eq. (5).

---

> > ### Author Response · Authors · 2022-11-22
> > **Response to Area Chair RUf5: part 2**
> >
> > 3. CDGNode:
> >
> > **Summary:** Same as CGNode except that the architecture is modified to decouple the computations of internal and body forces. The raw node and edge inputs are the same. However, since we learn two representations per node, global and local, these inputs are used in the following manner:
> > * **Input for global node representations:** position ($q$) and velocity ($\dot{q}$); these features are not involved in message passing.
> > * **Input for local node representations:** type ($t$).
> > * **Input for local edge representations:** $w_{ij}=(q_i-q_j)$, i.e., same as in GNode. Note that node local input and edge inputs are involved in message passing.
> > * **Node output:** Same as in CGNode, but the local node output and the global node output are concatenated and passed through an MLP to obtain the total node force ($N_i$) as shown in Fig. 1. The intuition is that the local node output represents the internal forces (for instance, the spring forces) and the global node output represents the body forces (for instance, the gravitational force), which when combined gives the total force on each particle. This design thus enables more expressive power for the neural network.
> > * **Edge outputs:** Nil, same as in CGNode.
> > * **Architecture:** Fig. 1.
> > * **Loss function:** Same as in CGNode.
> > * **Assumed knowns:** Same as in CGNode.
> > * **Inductive bias:** Same as in CGNode + decoupling the computation of internal forces (which depend on the topology and is a function of the $w_{ij}$) and body forces (which typically depends on the position of the particle rather than the topology of the system, for instance, gravitational or electromagnetic force).
> > 4. MCGNode:
> >
> > **Summary:** Same as CDGNode except that the architecture is modified to enforce Newton's third law by incorporation of an additional MLP to predict the individual interaction forces at the edge level.
> > * **Node global input, node local input, edge input:** Same as in CDGNode.
> > * **Edge output:** Force $f_{ij}$ on particle $i$ due to the interaction from the neighboring particle $j$.
> > * **Node output:** Total force $N_i$ for each particle given by $N_i = N_{gi} + \sum_{j \in \mathcal{N}\_i}f_{ij}$, where $\mathcal{N}\_i$ represents of the set of all neighbors of $i$ and $N_{gi}$ represents the force due to external fields obtained by passing the global node output through an MLP.
> > * **Architecture:** Fig. 8 with a difference that there are two MLPs, one to compute $N_{gi}$ which takes as input the global node representation and a second that takes the edge representation as input to predict the $f_{ij}$. We will include a separate figure to explicitly differentiate between CDGNode and MCGNode.
> > * **Loss function:** Same as in CGNode
> > * **Assumed knowns:** same as in CGNode.
> > * **Inductive bias:** Same as in CDGNode + Newton's third law, that is, the internal forces between the particles are equal and opposite.
> >
> > Although these points are included in the manuscript, we will consolidate the above information as an Appendix for easy understanding of readers.
> >
> > **Some questions about Figs. 1 and 8:
> > Is Figure 8 correct? I think GNODE is supposed to output predicted accelerations.**
> >
> > *Response:* We thank the area chair for raising this subtle but important point. As outlined earlier, GNode outputs the acceleration (this is clarified in the text as well). Note that predicting force and acceleration are equivalent problems in GNode since the acceleration is a simple scaling of the force. However, this is not so for the other models, namely, CGNode, CDGNode and MCGNode. In those cases, the graph outputs the force as explained above. Thus, while Fig. 8 mirrors the CGNode exactly, for GNode, the final output is indeed acceleration. We will address this by having two separate diagrams for GNode and CGNode.
> >
> > Note that, in general, due to the minor architectural variations we were unable to make one figure that is representative of all the architectures. To address this, we now propose to incorporate separate architecture diagrams for all the four models in the Appendix. We will clarify this point in the final version of the manuscript.
> >
> > **In Fig. 1, it is unclear which parts belong to CDGNODE and which to MCGNODE.**
> >
> > *Response:* Figure 1 represents CDGNode architecture with both local and global node features, the output of which is added to obtain the total nodal force. As clarified earlier, MCGNode employs an additional MLP to compute the internal interactions with the rest of the architecture being identical to CDGNode. To address this, we will add a separate figure for the MCGNode architecture.

---

> > > ### Author Response · Authors · 2022-11-22
> > > **Response to Area Chair RUf5: part 3**
> > >
> > > **Why do the topologies of the "graph representations of physical system" and the GNN differ?**
> > > *Response:* Indeed the topologies of the input "graph representations of physical system" and the GNN are identical. In Fig. 1, we show only the one-layer message passing  with respect to a single node, which in this case is the top left node (yellow node) in the input graph. Hence, only its one-hop subgraph is shown. The bars next to the nodes correspond to the representations being learned.
> > >
> > >
> > > Altogether, the summary of proposed changes to clarify the above questions are as follows:
> > > 1. to include architecture diagrams for MCGNode and GNode
> > > 2. to consolidate the above discussion outlining the differences between various models in the Appendix.
> > >
> > > We hope that we have addressed all the comments raised by the area chair. If there are any additional comments, we would be happy to further engage in a discussion.

---

### Decision · Program_Chairs · 2023-01-20

**Decision:**

Accept: poster

**Justification For Why Not Higher Score:**

relatively small contribution

**Justification For Why Not Lower Score:**

The models seem reasonable and perform well.

**Metareview: Summary, Strengths And Weaknesses:**

The paper presents a graph-based neural ODE model which can be used to learn the time evolution of a physical system. The dynamics function is modeled with a graph neural network which allows applying a trained model to a different number of objects at test time. The authors propose several versions of the model:
- GNode: Each node of the GNN takes an object's position and velocity as inputs and the GNN predicts the acceleration of each object.
- CGNode: GNode + taking into account known constraints (holonomic or Pfaffian) based on the topology and nature of the system.
- CDGNode: CGNode + explicitly modeling external forces (for instance, gravitational or electromagnetic forces, which depend on the positions of objects rather than the topology of the system).
- MCGNode: CDGNode + Newton's third law (the internal forces between the particles are equal and opposite).

The experimental results suggest that introducing the inductive biases of the constraints, internal-external forces, and conservation of Newton’s third law improves the accuracy of the trained model.

During the discussion, the following concerns were raised by the reviewers:
- It was not always clear in the presentation what was assumed known and what was learned from data. The authors provided a good clarification of that.
- Trying more complex system. The authors provided experimental results with a 3D solid system subjected to isotropic compression.
- The differences between the proposed method and LGN/HGN are not clear. The authors responded that MCGnode incorporates three different inductive biases: Newton's third law (Sec 3.4), decoupling of internal and external forces (Sec 3.3) and Decoupling dynamics and constraints (Sec 3.2).
- Include more comparison models that can use physical knowledge (e.g. Symplectic ODE-Net) and study the robustness against noise. The authors added Symplectic ODE-Net as a baseline and included experiments studying robustness to noise.

The reviewers were generally convinced that the paper contains novel contributions and they were satisfied with the quality of the paper. The authors are encouraged to include the extra clarifications and experimental results produced during the discussion in the paper.

**Note From Pc:**

if the above contains the word "oral" or "spotlight" please see: "oral" presentation means -> notable-top-5% and "spotlight" means -> notable-top-25%. As stated in our emails, we are disassociating presentation type from AC recommendations